# Treatment of atherosclerosis by macrophage-biomimetic nanoparticles via targeted pharmacotherapy and sequestration of proinflammatory cytokines

Cheng Gao [1,2], Qiaoxian Huang[1,2], Conghui Liu [1,2], Cheryl H. T. Kwong[1], Ludan Yue[1], Jian-Bo Wan[1], Simon M. Y. Lee[1 ✉] & Ruibing Wang [1 ✉]

Vascular disease remains the leading cause of death and disability, the etiology of which often involves atherosclerosis. The current treatment of atherosclerosis by pharmacotherapy has limited therapeutic efficacy. Here we report a biomimetic drug delivery system derived from macrophage membrane coated ROS-responsive nanoparticles (NPs). The macrophage membrane not only avoids the clearance of NPs from the reticuloendothelial system, but also leads NPs to the inflammatory tissues, where the ROS-responsiveness of NPs enables specific payload release. Moreover, the macrophage membrane sequesters proinflammatory cytokines to suppress local inflammation. The synergistic effects of pharmacotherapy and inflammatory cytokines sequestration from such a biomimetic drug delivery system lead to improved therapeutic efficacy in atherosclerosis. Comparison to macrophage internalized with ROS-responsive NPs, as a live-cell based drug delivery system for treatment of atherosclerosis, suggests that cell membrane coated drug delivery approach is likely more suitable for dealing with an inflammatory disease than the live-cell approach.

[1] State Key Laboratory of Quality Research in Chinese Medicine, Institute of Chinese Medical Sciences, University of Macau, Taipa, Macao, China. [2]These authors contributed equally; Cheng Gao, Qiaoxian Huang, Conghui Liu. ✉email: simonlee@umac.mo; rwang@umac.mo

Cardiovascular disease remains the leading cause of death globally[1]. The disease generally involves atherosclerosis. Although the atherosclerosis process is not well understood, it is often initiated by the endothelial layers' dysfunction, which accumulates oxidized form of low-density lipoprotein (LDL) in the intimal layer and leads to the local inflammation where reactive oxygen species (ROS) are overproduced[2,3]. The inflammatory areas recruit monocytes and differentiate into macrophages. Upon ingestion of LDL, macrophages would die or even lead to cellular rupture, providing positive feedback to recruit additional immune cells to these areas[4,5]. Subsequently, the inflammation leads to formation of atheromatous plaques in the arterial tunica intima. Treatment of established plaques may include medications to lower cholesterol, such as statins, or medications that decrease clotting, such as aspirin[6]. However, only limited efficacy has been observed in clinics when the drugs are administered systemically, likely attributed to the rapid drug clearance and unsatisfactory accumulation at the arterial injury site[7]. Considering these issues, significant efforts have been devoted to the design of various nanomaterials, which could control payload release at the plaque sites in response to the highly produced ROS or the acidic environment, or may enhance targeted drug delivery via a targeting ligand, or extend systemic circulation[8,9].

However, once administered into the body, nanomaterials encounter extremely complexed physiological environments and defense (immune) system that is actively recognizing and clearing matters that are foreign to our human body[10,11]. Thus, very few NPs-based formulations coated with targeting ligands have passed phase III clinical trials. One of the major hurdles for the clinical applications of NPs is that most NPs are taken and removed by the reticuloendothelial system (RES) before reaching the target tissues[12,13]. Very recently, Chan and co-workers prepared trastuzumab and folic acid coated gold and silica NPs, and quantified their cancer cell targeting efficiencies[14]. Their results demonstrated that only 0.7% i.v. administered NPs reached the tumor site and only 0.0014% reached inside targeted cancer cells, suggesting that NPs, even tagged with targeting ligands, might not achieve the expected delivery efficiency.

As one of the most fundamental units of life, cells get along well with its surrounding environments and may accumulate in some specific microenvironments depending on their nature[15]. Biomimetic drug delivery systems, especially cell membrane coated NPs, have attracted rapidly increasing attentions[16,17]. For instance, Tasciotti et al. reported the first macrophage membrane (leukocyte membranes) coated nanoparticles that enhanced circulation time and improved accumulation in the tumor[18]. Zhang et al. recently reported dual-cell membrane-coated NPs from the fused membranes originated from both red blood cells (RBC) and platelets, and neutrophil membrane coated nanoparticles to alleviate inflammatory arthritis[19,20]. For the treatment of cardiovascular disease, platelet membrane coated nanoparticles were firstly reported to show enhanced therapeutic effects against coronary restenosis[21]. Similarly, Wang and coworkers developed RBC membrane coated nanocomplexes to minimize macrophage-mediated phagocytosis in the blood and enhance accumulation of nanoparticles in the established atherosclerotic plaques for improved atherosclerosis management[22]. In spite of the significantly reduced RES clearance and improved payload delivery, none of these previous carriers may specifically release payload in the disease sites. Considering the accumulation of macrophage and overproduction of ROS during the development of atherosclerosis[23], herein we developed a delivery system derived from macrophage membrane coated ROS-responsive NPs (MM-NPs) for the treatment of atherosclerosis. The macrophage membrane not only may improve targeted delivery of NPs and payload to the lesion site, but also may act as a scavenger for proinflammatory factors. Meanwhile, live cells have also recently emerged as potential drug carriers with various advantages[24]. For instance, neutrophils were applied as carriers for liposomes loaded with paclitaxel to fight against postoperative glioma recurrence, and haematopoietic stem cells were used for the delivery of antibodies to augment antileukemia efficacy[25,26]. As the inflammatory response of macrophage is not only related to the proteins in cell membrane, but also involves some signal pathways inside cells[27,28], we thought that ROS-responsive NPs internalized inside macrophage (NPs/MAs) might as well have a high targeting efficiency and excellent therapeutic efficacy on atherosclerosis. Thus, a thorough comparative study between MM-NPs and NPs/MAs was conducted for the treatment of atherosclerosis in vivo.

In our experimental design, atorvastatin (AT) is selected as the model drug and positive control. The attenuation effects of AT loaded ROS-NPs (AT-NPs) and macrophage membrane coated AT-NPs (MM-AT-NPs) on lipopolysaccharide (LPS) and oxLDL induced macrophage inflammation and foam cell formation are, respectively, demonstrated. The escape behavior of MM-NPs from macrophage, as well as the intracellular drug release in ROS overproduced cells are exhibited in vitro. Biocompatibility of MM-NPs and macrophage is respectively confirmed in vivo. In an atherosclerotic mouse model, the targeting efficiency and therapeutic efficacy of MM-AT-NPs and AT-NPs internalized inside macrophages (AT-NPs/MAs) are systemically evaluated and compared against those of AT alone. Finally, the underlying therapeutic mechanisms are thoroughly investigated at a molecular level, and very interestingly, MM-AT-NPs exhibit hints of moderately better therapeutic efficacy than AT-NPs/MAs to treat the inflammatory disease.

## Results

**Preparation of ROS responsive NPs and MM-NPs.** ROS responsive NPs were prepared via self-assembly of amphiphilic oxidation-sensitive chitosan oligosaccharide (Oxi-COS). The synthetic procedure of Oxi-COS was shown in Supplementary Fig. 1, and the successful chemical conjugation was confirmed by $^1H$ NMR (Supplementary Fig. 2). In an aqueous solution, the phenylboronic acid pinacol ester serves as the hydrophobic side chain, which would aggregate to form a hydrophobic core, and the hydrophilic backbone of COS would spread outside around the hydrophobic core to form NPs with a micelle structure. Subsequently, NPs loaded with hydrophobic AT (AT-NPs), with a drug encapsulation efficiency (DEE) of 48.3% and a drug loading content (DLC) of 5.1%, were obtained by a simple self-assembly process (Fig. 1a). Transmission electron microscope (TEM) analysis showed that these NPs possessed roughly spherical morphology with a mean size of ca. 149 nm (Fig. 1b). It is well known that $H_2O_2$ oxidizes arylboronic esters[29,30], thus Oxi-COS would degrade into pinacol borate, COS, and p-hydroxymethylphenol in the presence of $H_2O_2$ (overproduced at inflammatory tissues). In order to confirm the ROS responsiveness, NPs were placed in phosphate-buffered saline (PBS) with different concentrations of $H_2O_2$ and their transmittance was measured at 500 nm at different time points. The concentrations of $H_2O_2$ used in the present study were based on previous literature papers that simulated the level of $H_2O_2$ in vitro for atherosclerotic lesion[9,31]. As shown in Fig. 1c, the particle solution in the presence of 1.00 mM $H_2O_2$ became completely transparent within 2 h. In contrast, the particles in PBS exhibited only moderate decrease over 4 h in absorbance due to partial precipitation of NPs. Accordingly, AT-NPs exhibited a high accumulative drug release rate up to ~80% in the presence of 1.00 mM $H_2O_2$, whereas only ~20% of drug was released without

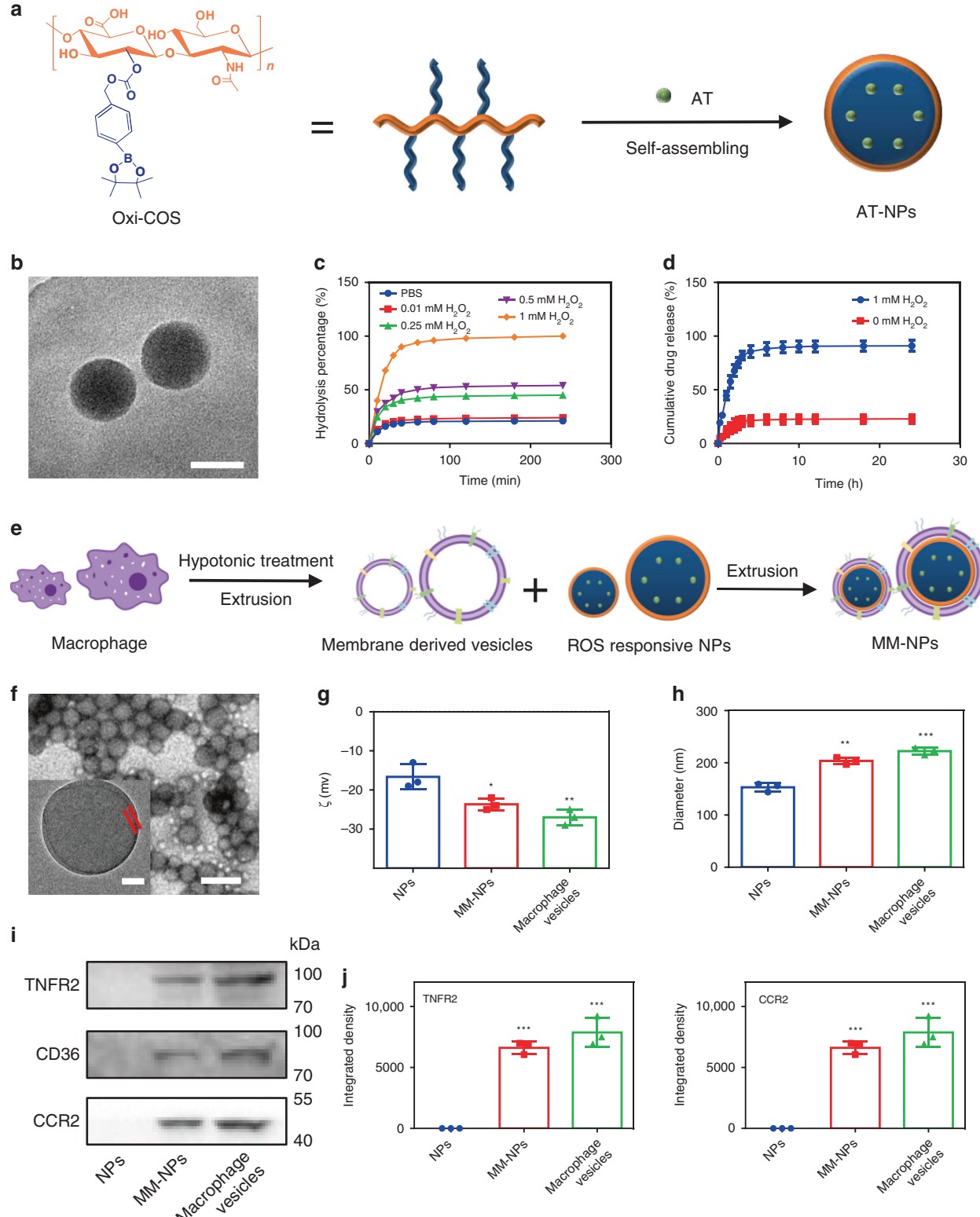

peroxide treatment (Fig. 1d), indicating the excellent ROS responsiveness of these NPs.

To synthesize MM-NPs, macrophage membrane derived from murine macrophage cell line (RAW264.7 cells) was coated on the surface of ROS responsive NPs via an extrusion method (Fig. 1e). As shown in Fig. 1f, MM-NPs exhibited a spherical core–shell structure under TEM, and each NP was wrapped with a single

layer of cell membrane, as the thickness of the wrapped layer was ~9 nm, in line with the thickness of cell membrane. Dynamic laser scattering (DLS) indicated that the zeta potential of MM-NPs was more negative than that of free NPs (Fig. 1g), but consistent with the zeta potential of macrophage surface. Furthermore, DLS analysis revealed that, after cell membrane coating, the diameter of nanoparticles was increased from ~204

**Fig. 1 Preparation and characterization of ROS responsive NPs and MM-NPs. a** Schematic illustration of the preparation of AT-NPs. **b** Representative TEM image of ROS responsive NPs. Scale bar: 100 nm. **c** Hydrolysis rate of NPs in PBS with different concentration of $H_2O_2$ (0, 0.01 mM, 0.25 mM, 0.50 mM and 1.00 mM). **d** In vitro drug release profile of AT-NPs w and w/o 1.00 mM of $H_2O_2$. **e** Schematic illustration of preparation of MM-NPs through an extrusion method. **f** Representative TEM image of MM-NPs. Scale bar: 500 nm. Inset: the amplified TEM image of a single MM-NP. Scale bar: 50 nm. **g, h** Zeta potentials and particle sizes of NPs and MM-NPs analyzed by DLS. **i** Characteristic protein bands of NPs, macrophage membrane derived vesicles, and MM-NPs resolved by Western blotting. **j** Quantitative analysis on the integrated density of TNFR2 and CCR2, measured in NPs (100 μL of suspensions), macrophages ($2.5 \times 10^7$ cells) and MM-NPs (100 μL of suspensions, 0.25 mg/mL protein content). In **g, h, j**, statistical comparison was made between MM-NPs and NPs, and between macrophage vesicles and NPs, respectively. All the experiments were repeated for three times ($n = 3$) and data was presented as mean ± s.d. Statistical analysis was conducted using one-way ANOVA. *$P \leq 0.05$, **$P \leq 0.01$, and ***$P \leq 0.001$. Source data are provided as a Source Data file.

to ~227 nm (Fig. 1h), again in line with the addition of bilayer of macrophage membrane. Moreover, MM-AT-NPs exhibited a high stability in PBS and serum (Supplementary Fig. 3a). In response to 1.00 mM of $H_2O_2$, the hydrolysis rate increased significantly (being very moderately slower than that of NPs without MM coating), and an accumulative drug release rate from MM-AT-NPs reached ~74.6% (Supplementary Fig. 3b, c), rather close to ~80% release rate from AT-NPs under the same conditions, indicating that the macrophage membrane coating did not significantly affect the responsiveness of the NPs. Furthermore, Western blot analysis confirmed the presence of key membrane antigens on the surfaces of both macrophage membrane and MM-NPs, such as tumor necrosis factor receptor receptor 2 (TNFR2), CD36 (a receptor for oxLDL), and CCR2 (a receptor for monocyte chemoattractant protein-1 (MCP-1)), suggesting that the macrophage membrane on MM-NPs was comparable to those on macrophage (Fig. 1i, j).

**Attenuation effects of MM-AT-NPs on inflammation in vitro.** Atherosclerosis is a chronic inflammatory disease and the formation of foam cells plays a key role in disease development[32,33]. Therefore, LPS-induced inflammation in macrophage and oxLDL-treated macrophage (foam cells formation) were selected as in vitro models for evaluating the therapeutic efficacy of MM-AT-NPs. We firstly evaluated the safety of blank NPs on HUVECs and RAW264.7 cells (Supplementary Fig. 4a, b). Nearly, no cytotoxicity was observed on both cell lines even at 1 mg mL$^{-1}$ of NPs. Subsequently, we constructed inflammatory macrophage and foam cells through LPS and oxLDL treatments of macrophage, respectively. After treatment with LPS (10 ng mL$^{-1}$) or oxLDL (20 μg mL$^{-1}$) for 24 h, an obvious cytotoxicity was observed (Fig. 2a, Supplementary Fig. 4c). When macrophage was co-treated with AT, AT-NPs, or MM-AT-NPs, respectively, cellular viability improved in a dose-dependent manner, although no significant difference was observed among these groups.

Subsequently, the ROS levels of macrophage treated with LPS or oxLDL were measured. The ROS overproduced in these treated cells (Fig. 2b, c, Supplementary Fig. 4d–f) must have led to the release of AT from AT-NPs or MM-AT-NPs, resulting in similar therapeutic efficacy. Furthermore, AT-NPs and MM-AT-NPs decreased the NO production on LPS induced macrophage in a way similar to AT (Fig. 2d). In addition, we observed that these AT formulations also inhibited the cellular apoptosis on LPS induced macrophages (Supplementary Fig. 4g, h), attenuated the oxLDL-induced immature dendritic cells-like morphological changes of macrophage (Supplementary Fig. 5), and decreased the cellular uptake of oxLDL (Supplementary Fig. 6). Collectively, these data demonstrated the ROS responsiveness of AT-NPs and MM-AT-NPs in vitro, as well as their attenuation effects on LPS induced inflammation and oxLDL induced foam cell formation from macrophage, suggesting a significant potential of these biomaterials for atherosclerosis therapy.

**Cellular uptake and intracellular drug release.** The current drug delivery systems are often cleared out by mononuclear phagocyte system before reaching a targeted site[34,35]. Therefore, we investigated the uptake efficiency of MM-NPs by macrophage. Cyanine 5 NHS ester (Cy5), a red fluorescence dye, was encapsulated into the cores of NPs (Cy5-NPs) and MM-NPs (MM-Cy5-NPs), respectively, through hydrophobic interactions between Cy5 and with the lipid core of NPs, in order to track the cellular uptake process. As shown in Fig. 2e, strong red fluorescence was observed in macrophage incubated with Cy5-NPs for 4 h, suggesting that Cy5-NPs were significantly taken up by macrophage. In a dramatic contrast, nearly no fluorescence was observed in macrophage incubated with MM-Cy5-NPs, indicating that drug delivery systems coated by macrophage membrane may provide an effective strategy to escape macrophage clearance. When macrophage was treated with LPS or oxLDL to induce inflammation or to form foam cells, a strong red fluorescence was observed in these cells incubated with either MM-Cy5-NPs (Fig. 2e) or Cy5-NPs (Supplementary Fig. 7), indicating that inflammatory macrophage had significantly high uptake efficiency toward both delivery systems without discrimination against MM-NPs.

To examine the intracellular drug release behavior, Nile red (NR), a lipophilic stain with little fluorescence in a polar solution but strong red fluorescence in a lipid environment[36], was loaded into MM-NPs (MM-NR-NPs), and incubated with LPS- and oxLDL-treated macrophage, respectively. As shown in Fig. 2f, the red fluorescence of NR increased in an incubation-time dependent manner, indicating the NR was released in LPS and oxLDL treated, inflammatory macrophages, respectively (Fig. 2b, Supplementary Fig. 4e). Taken together, the escape of MM-NPs from macrophages, their efficient cellular uptake and responsive payload release in inflammatory macrophages and foam cells suggested that the ROS-responsive NPs coated with macrophage membrane may provide a potential targeted delivery platform for improved therapy of atherosclerosis.

**Biocompatibility of MM-NPs in vivo.** To systemically evaluate the biocompatibility of macrophage membrane coated ROS-responsive NPs (MM-NPs), two groups of C57BL/6 mice were i.v. injected with saline and MM-NPs, respectively, for half a month with a high dose of 100 mg kg$^{-1}$ MM-NPs every four days (for a total of four injections). The histological studies on the heart, livers, spleen, lungs, and kidneys (Supplementary Fig. 8a) showed no obvious damage in mice treat with MM-NPs. In addition, as shown in Supplementary Fig. 8b, c, the liver function biomarkers (alanine transaminase (ALT) and aspartate aminotransferase (AST)) and kidney function biomarkers (blood urea nitrogen (BUN) and uric acid (UA)), and the inflammatory cytokine levels (TNF-$\alpha$, IL-6, and IL-1$\beta$) in the serum of mice treated with MM-NPs were comparable to those of the saline-treated group. Furthermore, the changes of bodyweight of MM-NPs treated mice during the 15-day follow-up were similar to those of the saline-

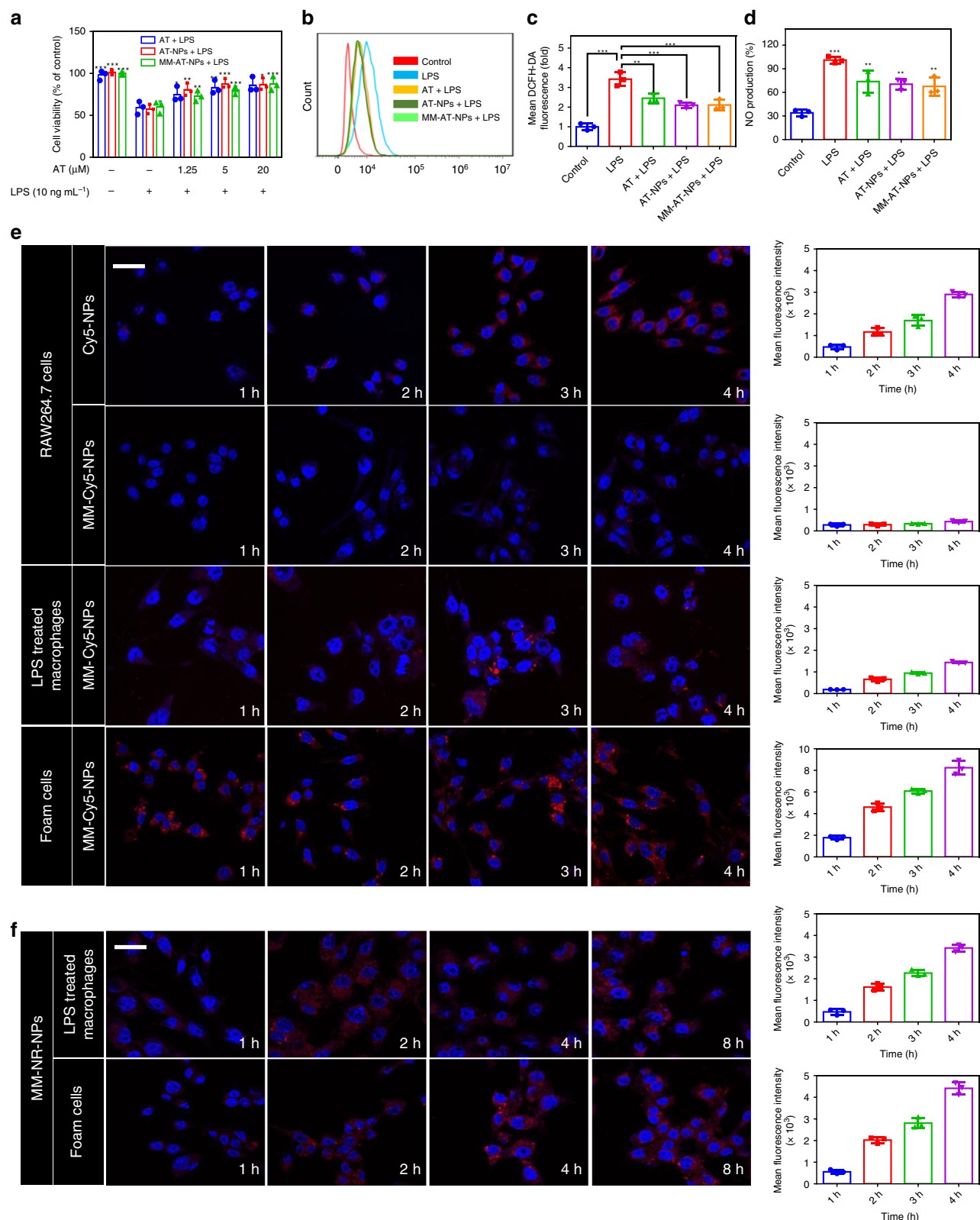

treated control group (Supplementary Fig. 8d). Collectively, this set of data support the good biocompatibility of MM-NPs in vivo.

**Targeted delivery in atherosclerotic mouse.** In the developmental process of atherosclerosis, atheromatous plaques secrete inflammatory cytokines and chemokines, leading to the recruitment of macrophages[37,38]. Taking this into account, live macrophage might be an ideal drug delivery system for AT-NPs due to their immune response and proactive migration to inflammatory sites[39]. In this study, RAW264.7 cells were selected as live cell based carriers for AT-NPs. Thus, we prepared AT-NPs/MAs (Fig. 3a) to investigate their therapeutic efficacy against atherosclerosis in mice, and to compare with that of

**Fig. 2 Attenuation effects on LPS-induced inflammation and oxLDL-induced foam cells formation by MM-AT-NPs. a** Viability of RAW264.7 co-treated with LPS (10 ng mL$^{-1}$) and AT, AT-NPs, and MM-AT-NPs, respectively, at 0, 1.25, 5, and 20 μM AT. **b, c** Intracellular ROS levels (by flow cytometry analysis) in RAW264.7 cells treated with LPS (400 ng mL$^{-1}$), w or w/o AT, AT-NPs, or MM-AT-NPs, respectively at 0.4 mM AT for 24 h. **d** NO production of RAW264.7 cells treated with LPS (100 ng mL$^{-1}$), w or w/o AT, AT-NPs or MM-AT-NPs, respectively at 0.1 mM AT for 24 h. **e** Cellular uptake (including quantitative analysis) of Cy5-NPs and MM-Cy5-NPs by RAW264.7 cells treated with LPS and oxLDL, respectively. Scale bar: 50 μm. **f** Intracellular payload release (including quantitative analysis) of MM-NR-NPs in LPS- and oxLDL-treated macrophage. Scale bar: 50 μm. The experiments were repeated for three times ($n = 3$) and data were presented as mean ± s.d. Statistical analysis for cell viability was performed using two-way ANOVA. Analysis for mean DCFH-DA fluorescence, NO production and apoptosis rate were conducted using one-way ANOVA. *$P \leq 0.05$, **$P \leq 0.01$, and ***$P \leq 0.001$. Source data are provided as a Source Data file.

MM-AT-NPs. Firstly, we investigated the systemic biocompatibility of RAW264.7 cells in 6-week-old female ApoE$^{-/-}$ mouse. After i.v. administration with different number of macrophages into mice once every four days continuously for half a month, it was found that the body weight (Supplementary Fig. 9a), and counts of immune-associated cells including monocyte, lymphocyte and neutrophil in the blood (Supplementary Fig. 9b) of the treated mice were similar to those of the mice in the control group. In addition, the histological analysis of the main organs (including the liver, lungs, kidneys, and spleen) of the mice treated by macrophage indicated no toxicity (Supplementary Fig. 9c). These results demonstrated that no obvious immunotoxicity was caused in mice administered with external macrophage, even at the highest dosage (10$^9$ macrophages kg$^{-1}$). Upon internalization of a dye-loaded NPs, Cy5-NPs, by macrophage, red fluorescence was mainly distributed in cytoplasm (Supplementary Fig. 10a). The DEE of AT in the AT-NPs/MAs (5 × 10$^7$ cells) was determined by HPLC to be ~14.7%, and more importantly, AT-NPs showed negligible cytotoxicity to macrophage even at a highest concentration of 20 μM used in our study (Supplementary Fig. 10b).

To study the targeting capability of MM-NPs and NPs/MAs to atheromatous plaques, cyanine 7.5 NHS ester (Cy7.5), a synthetic dye with near infrared emission (excitation/emission 788/808 nm), were employed for preparing Cy7.5-loaded NPs (Cy7.5-NPs), macrophage membrane coated Cy7.5-NPs (MM-Cy7.5-NPs), and Cy7.5-NPs internalized inside macrophage (Cy7.5-NPs/MAs). As was previously reported, atherosclerotic plaques begin to develop in ApoE$^{-/-}$ mouse after being fed with high-fat diet for 1 month, during which process macrophages migrate to plaques in large numbers[40]. Thus, 6-week-old female ApoE$^{-/-}$ mice having received high-fat diet for 1 month were employed for our investigation of the targeted delivery. To determine a suitable imaging time point for plaque targeting studies, in vitro pharmacokinetic studies of Cy7.5-NPs, MM-Cy7.5-NPs, and Cy7.5-NPs/MAs were determined by IVIS (in vivo imaging spectrum) system by testing on blood samples taken from mice at predetermined time points after i.v. administration of each of these formulations. As shown in Supplementary Fig. 11, the fluorescence intensities became significantly weaker after administration of these formulations for longer than 6 h, and got almost completely cleared after 12 h. The circulation half-life ($t_{1/2}$) of MM-Cy7.5-NPs, 9.82 h, was much longer than Cy7.5-NPs ($t_{1/2}$ = 5.43 h), and Cy7.5-NPs/MAs ($t_{1/2}$ = 13.32 h) exhibited the longest circulation time among all groups. Therefore, at 6 h after i.v. injection of different formulations (free Cy7.5, Cy7.5-NPs, MM-Cy7.5-NPs, and Cy7.5-NPs/MAs, respectively) with the same dosage of Cy7.5 (2 mg kg$^{-1}$) into mice, the aorta harvested from all treated groups of mice exhibited different levels of fluorescence (Fig. 3b), whereas the blank control groups exhibited no fluorescence (Fig. 3c). Among which, Cy7.5-NPs treated group showed slightly stronger fluorescence than that of free Cy7.5 treated group in the aorta tissues, likely due to the specific release of Cy7.5-NPs in response to overproduced ROS in the

inflammatory aorta. In particular, strong fluorescence was observed in the aorta tissues isolated from both MM-Cy7.5-NPs and Cy7.5-NPs/MAs treated groups, likely due to their inherent immune propensity for inflammation. Of note, Cy7.5-NPs/MAs even exhibited a moderately higher accumulation rate in aorta tissue than MM-Cy7.5-NPs. The enhanced targeting efficiency of Cy7.5-NPs/MAs was likely attributed to active recruitment of macrophages in the development process of atherosclerosis.

**Therapeutic efficacy in atherosclerotic mouse.** We next examined the therapeutic efficacy of different formulations (saline, AT, AT-NPs, MM-AT-NPs, and AT-NPs/MAs) against atherosclerotic development. After receiving high-fat diet for 1 month, 6-week-old female ApoE$^{-/-}$ mice were randomly and investigator-blindly divided into 5 groups ($n = 10$ in each group from two batches of studies), and intravenously administered with different formulations, respectively, once a week, in combination with high-fat food for another 2 months (Fig. 3d). At endpoint of the experiment, the aorta was collected and stained by ORO and the resultant red region indicated plaque area (Fig. 3e). As shown in Fig. 3f, the saline-treated group showed a highest plaque area of ~20% of the total aorta tissue area, determined by using en face analysis of lesions on the intimal surface of the aorta[41]. The free drug AT moderately reduced the plaque area down to ~15%. Benefiting from the ROS responsiveness, AT-NPs exhibited slightly improved therapeutic efficacy in comparison to that of free AT. To our surprise, the aorta of mice treated with AT-NPs/MAs, which we expected to have excellent therapeutic efficacy against inflammatory plaques due to their best payload accumulation in the targeted site, exhibited similar plaque area (~14%) with that of the aorta of mice treated with AT-NPs (Fig. 3e, f). Very interestingly, MM-AT-NPs significantly decreased plaque area down to ~8% of the total aorta tissue area, showing moderate, yet obvious improvement in comparison with AT-NPs/MAs. Subsequently, we conducted ORO staining with sequential 10 cryosections at 100 μm intervals from the aorta tissues (Supplementary Fig. 12). In MM-AT-NPs treated group, an obvious decrease of plaque area was observed in the aorta root, in comparison to other groups.

Furthermore, histological and immunohistochemical analysis was conducted on the atherosclerotic plaques from the aorta root. As shown in Fig. 4a, hematoxylin and eosin (H&E) staining on the aorta root showed that the plaques from the saline-treated group (the control) and AT-treated group were largely necrotic cores. In contrast, the areas of plaques and necrotic cores were significantly reduced in the MM-AT-NPs treated group. Separate staining with anti-CD14 antibody and anti-matrix metalloproteinase-9 (MMP-9) antibody, respectively, exhibited that MM-AT-NPs effectively reduced the number of monocytes and the expression of MMP-9 in plaques of the aorta arch. As the total area of necrotic cores and monocyte filtration were positively related to plaque development and disease severity, MM-AT-NPs effectively prevented the atherosclerotic process, moderately

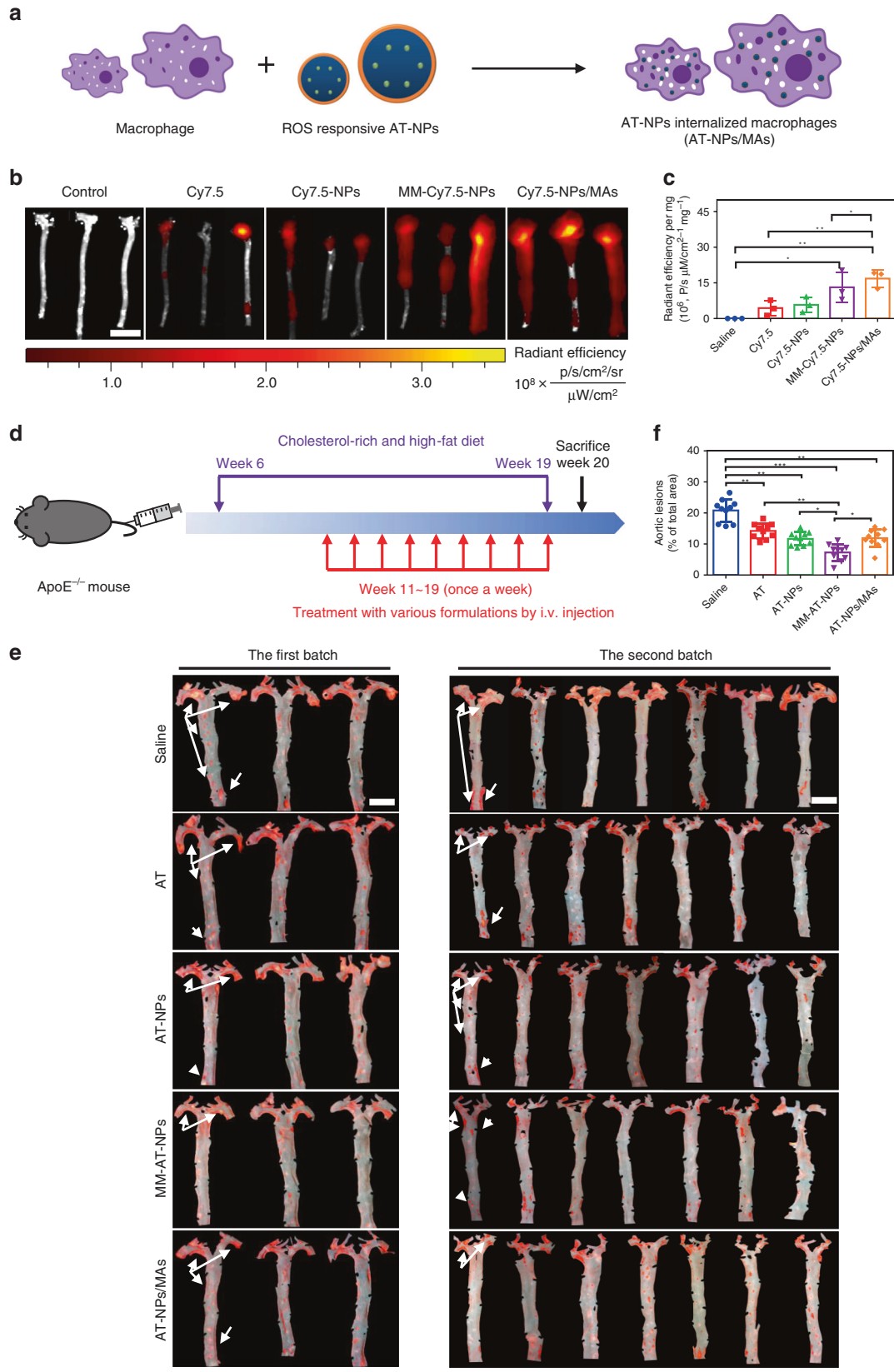

better than AT-NPs and AT-NPs/MAs. Furthermore, the level of collagen around plaques in MM-AT-NPs treated groups was determined by Masson's trichrome staining, showing a high collagen concentration and an enhanced fibrous cap thickness. It was previously reported that the proliferation of vascular smooth

muscle cells (VSMC) was involved in the inhibition of atherogenesis, and the staining of plagues with α-smooth muscle actin (α-SMA) was performed to measure the accumulation of VSMC. As shown in Fig. 4a, MM-AT-NPs treated group exhibited the highest accumulation rate of VSMC. Thus, both

**Fig. 3 Targeting efficiency and therapeutic efficacy of MM-AT-NPs and AT-NPs/MAs. a** Schematic illustration of preparation of AT-NPs/MAs. **b, c** Ex vivo fluorescence bio-imaging and quantitative analysis of Cy7.5 fluorescent signal in aorta tissues. ApoE$^{-/-}$ mice fed with high fat food for 1 month were i.v. administered with Cy7.5, Cy7.5-NPs, MM-Cy7.5-NPs, and Cy7.5-NPs/MAs, respectively. $n = 3$ aorta tissues from different mice. Scale bar: 15 mm. **d** Schematic illustration of atherosclerotic mouse model development and treatment with various formulations (AT, AT-NPs, MM-AT-NPs, and AT-NPs/MAs). **e** ORO stained aorta tissues collected from atherosclerotic mice after treatment with various formulations (AT, AT-NPs, MM-AT-NPs, and AT-NPs/MAs) at equivalent dosage of 2 mg kg$^{-1}$ AT per week. $n = 10$ aorta tissues from different mice. Scale bar: 5 mm. **f** Quantitative analysis of lesion area in aorta tissues ($n = 10$). All data were presented as mean ± s.d. Statistical analysis was conducted using one-way ANOVA. *$P \leq 0.05$, **$P \leq 0.01$, and ***$P \leq 0.001$. Source data are provided as a Source Data file.

the enhanced collagen concentration around plaques and increased accumulation of VSMC suggested that MM-AT-NPs stabilized the plaques from further development, and inhibited atherogenesis. Similarly, CD31 immunohistochemical study showed that MM-AT-NPs reduced the number of perivascular CD31(+) neovessels, and KI67 staining analysis results exhibited that MM-AT-NPs effectively inhibited the endothelial proliferation (Supplementary Fig. 13).

In addition, as shown in Fig. 4b–f, the lowest expression of major pro-inflammatory cytokines in both the aorta tissues (TNF-$\alpha$, IL-1$\beta$, and IL-6) and the blood serums (TNF-$\alpha$ and IL-6) were observed in the MM-AT-NPs treated group of ApoE$^{-/-}$ mice with atherosclerosis, when compared with those from all other formulations-treated groups. In line with these observations, MM-AT-NPs treated group also displayed the lowest level of oxLDL (measured in the phospholipid form, oxPL-LDL, by an assay kit) in the aorta tissue (Fig. 4g). Therefore, these data supported that MM-AT-NPs effectively decreased the systemic inflammation as well as oxPL-LDL levels and local inflammation in the aorta. Furthermore, the total cholesterol (TC) level did not change obviously in serum of the mouse treated with MM-AT-NPs, while high density lipoprotein cholesterol (HDL-C) level was increased moderately in the serum of all treated groups. Meanwhile, non-HDL-C levels in the treated groups exhibited little changes when compared with the control group (Fig. 4h). In addition, as shown in Fig. 4i, these formulations had little influence on the changes of body weight of the treated mice. Collectively, a series of evidences suggested that MM-AT-NPs exhibited excellent therapeutic effects against atherosclerosis in mice and showed hints of better treatment effects than AT-NPs/MAs.

**Anti-atherosclerotic mechanism of MM-NPs.** To further investigate the mechanism responsible for in vivo atherosclerotic treatment of these formulations, dihydroethidium (DHE) staining was conducted on sections of the aorta root, aorta arch, and brachiocephalic artery collected from atherosclerotic mice to evaluate their ROS levels. As shown in Fig. 5a, bright red fluorescence was observed in the saline-treated group (the control group), indicating that a high level of ROS was produced in these aorta tissues. Moreover, the saline-treated group also showed the highest level of H$_2$O$_2$ (Supplementary Fig. 14a), revealing that oxidative stress was significantly increased in atherosclerotic mice. As was discussed in the previous section, NPs had a good ROS responsiveness in the presence of a high level of H$_2$O$_2$ (Fig. 1c, d), or overproduced ROS in LPS induced macrophage (Fig. 2b, c) and foam cell (Supplementary Fig. 4e). After i.v. injection with different formulations, these NPs may respond to over-produced ROS in the inflammatory plaques, and release AT, exhibiting their anti-atherosclerotic effects. In comparison to free AT, ROS-responsive release in the plaque site gave these NPs a clear advantage in atherosclerotic therapy. Thus, a weak fluorescence intensity and a low level of H$_2$O$_2$ were observed in the AT-NPs, MM-AT-NPs, and AT-NPs/MAs treated group.

Attributed to the presence of membrane antigens (e.g., TNFR2, CD36, and CCR2) on macrophage, we hypothesized that macrophage membranes from MM-NPs might sequester proinflammatory cytokines or chemokines (Fig. 1i, j), which have been shown to play prominent roles in the atherosclerotic process[4]. In fact, we discovered the lower concentrations of inflammatory cytokines in the plaques of mice treated with MM-AT-NPs, in comparison to those of AT-NPs/MAs-treated groups (Fig. 4b–f). Herein, we studied the interactions between MM-NPs and inflammatory cytokines or chemokines. The representative cytokines (TNF-$\alpha$ and IL-1$\beta$) were incubated with different doses of MM-NPs, and MM-NPs dose dependent sequestration profiles were observed for both cytokines (Fig. 5b). The IC$_{50}$ (half maximal inhibition concentration) values of MM-NPs were 1219 and 399.7 μg mL$^{-1}$, respectively for TNF-$\alpha$ and IL-1$\beta$ clearance. Meanwhile, it was reported that both MCP-1 and oxLDL contribute to the plaque formation[42,43]. As shown in Fig. 5b, MM-NPs exhibited a good binding affinity toward both MCP-1 and oxLDL in a dose dependent manner. IC$_{50}$ values were 281.6 and 2813 μg mL$^{-1}$, respectively for MCP-1 and oxLDL inhibition. In addition, the blood serums collected from atherosclerotic mice were incubated with different doses of MM-NPs and similar binding kinetics were obtained (Supplementary Fig. 14b–e). Thus, these results revealed that MM-NPs may sequester proinflammatory cytokines and chemokines. Subsequently, the interaction of RAW264.7 cells with MCP-1 and oxLDL was also investigated for comparative purpose. After treatment of macrophage with MCP-1 (20 ng mL$^{-1}$) or oxLDL (20 μg mL$^{-1}$) for 24 h, significant activation and inflammation of macrophage was detected, as evidenced by a high expression of TNF-$\alpha$, IL-1$\beta$, and IL-6 in the cellular medium (Fig. 5c, d). Interestingly, the levels of these cytokines and chemokines were decreased by the co-treatment of MM-NPs with a dose-dependent inhibitory effect. Furthermore, ORO staining was conducted on RAW264.7 cells after treatment with oxLDL in the absence and presence of various doses of MM-NPs (Fig. 5e). Large red area in macrophage was detected in the group treated with oxLDL only, suggesting a high uptake efficiency of oxLDL by macrophage. With co-treatment of MM-NPs, the red area was significantly reduced, with a MM-NPs-dose dependent manner (Fig. 5f), indicating that efficient binding between MM-NPs and oxLDL may have prevented the uptake of oxLDL. Taken together, our studies suggested that live macrophage may get easily activated by inflammatory cytokines or chemokines to release more cytokines, whereas macrophage membrane may effectively sequester key proinflammatory cytokines or chemokines to decrease their levels in the environment thereby decreasing local inflammation.

**Discussion**

In summary, our study described a macrophage-biomimetic drug delivery system, in which ROS responsive NPs were coated with macrophage membrane. Unlike the existing PEGylation and antibody-based targeted delivery strategies, the macrophage membrane coating strategy alone may lead to the escape of NPs from the RES and actively target inflammatory tissues. This

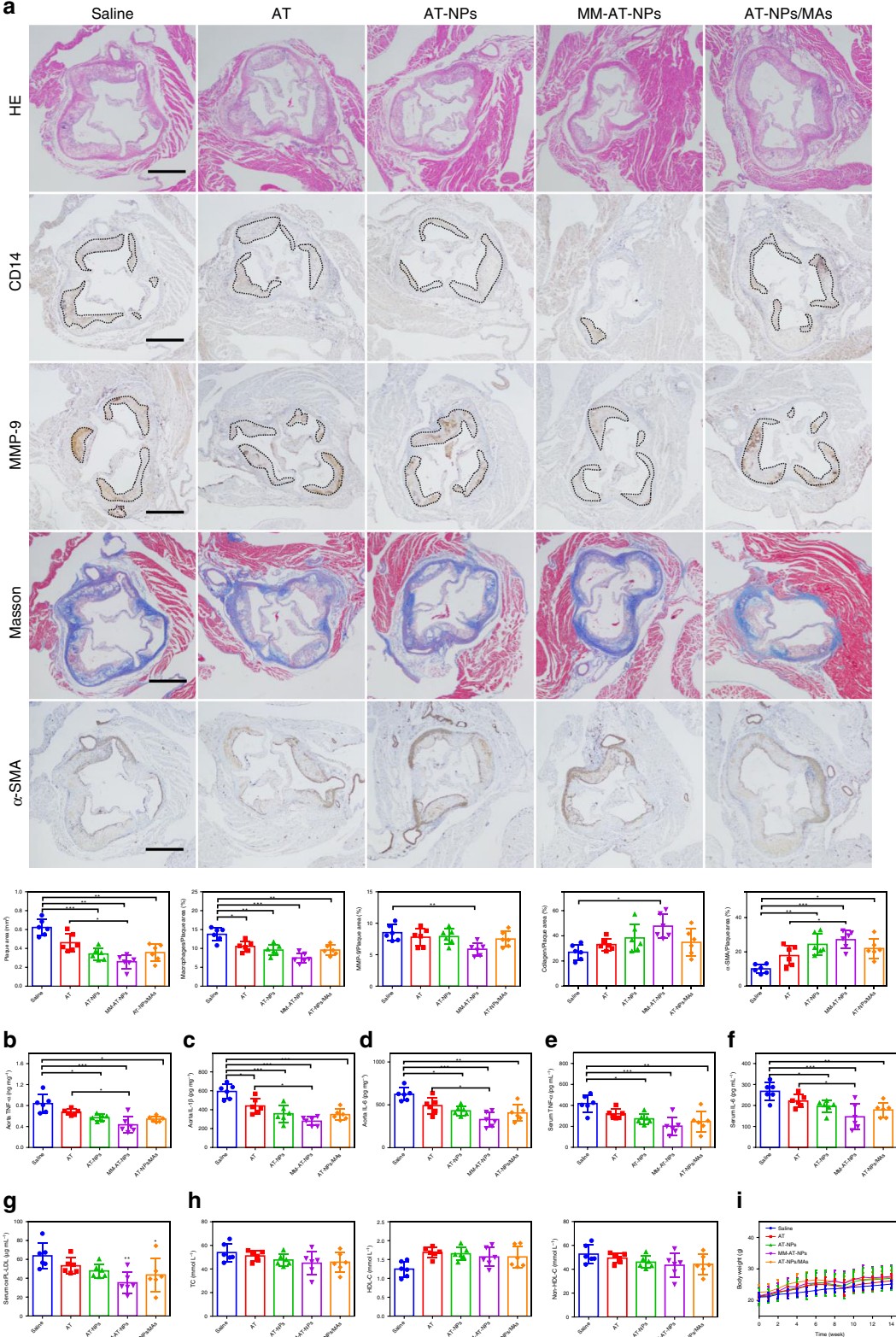

**Fig. 4 MM-AT-NPs ameliorated plaque and inflammation in an atherosclerotic mouse model. a** Representative photographs and quantitative analysis of aorta root sections stained by H&E, CD14 antibody, MMP-9 antibody, Masson's trichrome, and α-SMA antibody ($n = 6$). Scale bar: 500 μm. **b–d** The levels of TNF-α, IL-1β, and IL-6 in aorta tissues collected from atherosclerotic mice after treatment with various formulations (saline, AT, AT-NPs, MM-AT-NPs, and AT-NPs/MAs) at a dose of 2 mg kg$^{-1}$ AT per week ($n = 6$). **e–g** The levels of TNF-α, IL-6 and oxPL-LDL in blood serum ($n = 6$). **h** The levels of TC, HDL-C, and Non-HDL-C in the blood serum ($n = 6$). **i** The changes of body weight in atherosclerotic mice treated with various formulations (saline, AT, AT-NPs, MM-AT-NPs, and AT-NPs/MAs) ($n = 7$). All data were presented as mean ± s.d. Statistical analysis was conducted using one-way ANOVA. *$P \leq$ 0.05, **$P \leq 0.01$, and ***$P \leq 0.001$. Source data are provided as a Source Data file.

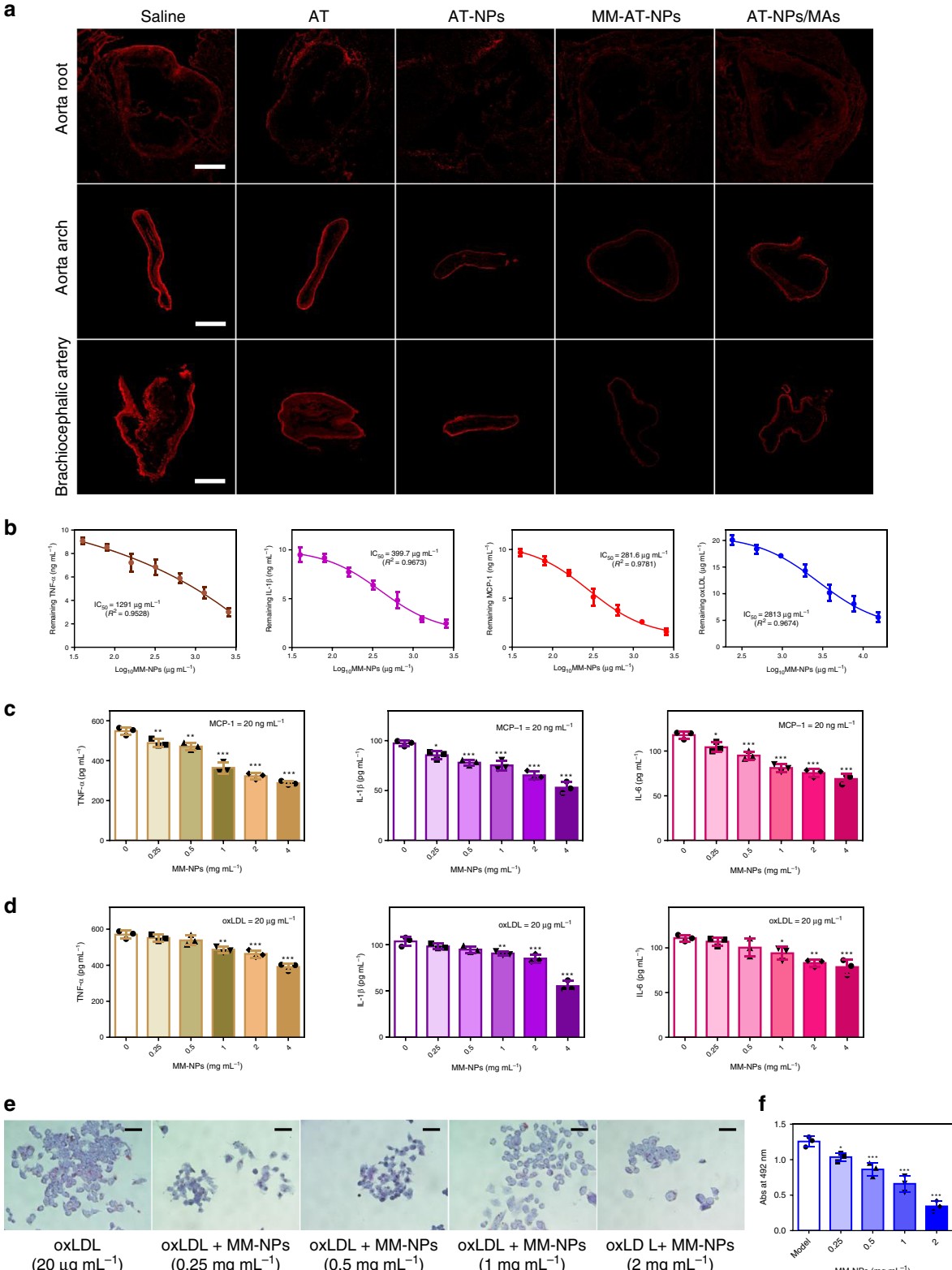

strategy provides a cellular function-driven, broad-spectrum functionalization strategy, and inflammatory tropism enabling targeted delivery without specific targeting molecules or complicated bioconjugation process. After accumulation at the targeted inflammatory tissues, the drugs loaded in NPs would be released in response to locally overproduced ROS, leading to effective pharmacotherapy. In addition, attributed to the presence of membrane antigens (e.g., TNFR2, CD36, and CCR2) on

macrophage, macrophage membrane coated NPs efficiently bind with, and sequester, multiple proinflammatory cytokines, and chemokines that play prominent roles in the atherosclerotic process, leading to inflammation neutralization. The combination of selective pharmacotherapy and sequestration of proinflammatory cytokines/chemokines offered by this platform exhibited significant synergistic therapies against atherosclerosis in mice.

**Fig. 5 Anti-atherosclerotic actions by MM-AT-NPs. a** DHE-stained sections of the aorta root, aorta arch and brachiocephalic artery, from atherosclerotic mice treated with various formulations (AT, AT-NPs, MM-AT-NPs, and AT-NPs/MAs) at a dose of 2 mg kg$^{-1}$ AT per week. Scale bar in aorta root and aorta arch: 400 μm. Scale bar in brachiocephalic artery: 800 μm. **b** Binding profiles of MM-NPs with TNF-α and IL-1β, MCP-1 (10 ng mL$^{-1}$ each), and oxLDL (20 μg mL$^{-1}$), with MM-NP varied from 0 to 4 mg mL$^{-1}$. Nonlinear regression fitting with inhibitory dose–response model (variable slope model) was employed to process data using Graphpad Prism 6. **c, d** MM-NP's dose-dependent inhibition of macrophage inflammation induced by MCP-1 and oxLDL, respectively, with MM-NP varied from 0 to 4 mg mL$^{-1}$. **e** Representative microscopic images of ORO stained RAW264.7 cells treated with oxLDL (20 μg mL$^{-1}$) and MM-NPs (0.25, 0.5, 1, and 2 mg mL$^{-1}$, respectively). Scale bar: 50 μm. **f** Quantified contents of ORO in foam cells derived from RAW264.7 cells. The experiments were conducted for three times independently. All data were presented as mean ± s.d. Statistical analysis was conducted using one-way ANOVA. *$P ≤ 0.05$, **$P ≤ 0.01$, and ***$P ≤ 0.001$. For DHE-stained aorta tissues, $n = 3$ aorta tissues from different mice. For binding capacity, inhibition of macrophage inflammation and ORO stained RAW264.7 cells, $n = 3$ independent experiments using same batch of MM-NPs. IC$_{50}$ was calculated by variable slope model using GraphPad Prism 6. Source data are provided as a Source Data file.

In addition, another biomimetic drug delivery system based on live macrophage was also developed in this study for a comparative purpose. Very interestingly, although live macrophage internalized with AT-NPs (AT-NPs/MAs) showed a higher accumulation rate in the plaque tissue than that of MM-AT-NPs, MM-AT-NPs exhibited hints of better therapeutic efficacy than AT-NPs/MAs. The accumulated macrophages might get activated by local inflammatory factors, leading to elevated inflammation and further recruitment of macrophages. In contrast, the macrophage membrane, without cellular activity, could not be activated. Of note, RAW264.7 cells were used as a model macrophage and a source of macrophage membranes in our proof of principle study. Potentially, bone marrow derived macrophages or a macrophage cell line with a high degree of similarity with plaque macrophages may serve as a better model of macrophage and source of macrophage membranes for biomimetic drug delivery systems to target atherosclerotic lesion.

The in vivo study indicated that MM-AT-NPs decreased atherosclerotic lesion area, from both the longitude and latitude artery staining. In particular, MM-AT-NPs degraded plaque vulnerability by increasing the ratio of collagen and α-SMA$^+$ VSMC content, and reducing CD14$^+$ macrophage infiltration and MMP-9 level. The released AT, in response to overproduced ROS, could optimize the lipid profiles and reduce atherosclerosis through decreasing the number of perivascular CD31$^+$ neovessels and inhibiting the KI67$^+$ endothelial proliferations. Meanwhile, MM-NPs scavenged proinflammatory cytokines and chemokines, leading to inflammation suppression, which was evident by the decreased levels of vascular NO, cytokines and chemokines. Therefore, MM-AT-NPs may have the potential to slow down, and to attenuate, plaque development in ApoE$^{-/-}$ mice in our study.

Mulder et al. previously reported that i.v. administration of statin-loaded reconstituted high-density lipoprotein (rHDL) nanoparticles helped target plaque macrophages in atherosclerotic lesions through the interactions between rHDL and plaque macrophage[44]. The local delivery of statins by rHDL to plaque tissue achieved obviously improved anti-inflammatory effects in comparison with free statin. Similarly, our study also leveraged the inherent anti-inflammatory effects of a statin drug, via a macrophage-biomimetic delivery platform. The macrophage membrane in our study not only acted as a targeted delivery vehicle due to the inflammatory homing effects, but also could sequester the inflammatory cytokines and chemokines to further reduce the inflammation level.

Besides atherosclerosis, this immune-cell biomimetic delivery strategy based on ROS responsive NPs may become a promising platform for other inflammatory diseases. The cell membrane could be derived from other immune cells, including neutrophil, T cells, and B cells[45]. In order to utilize the functions of different cells, multiple cell membrane coated NPs may also be developed for targeted and synergistic therapy. According to the different microenvironment of each specific disease, NPs may also be engineered to respond to pH, GSH, enzyme, light, or glucose to control the release of payload[46,47]. Therefore, this technology, combining immune cell membrane coating and stimuli-responsive NPs, may offer a variety of functionalities and advantages to treat various inflammatory diseases. Although the live macrophage carrier did not work well in our study, likely due to macrophage's inherent inflammatory activation, live cells may still become a potential intelligent drug delivery system for other diseases, especially for disease sites where is difficult for drugs to reach, such as brain[22] and bone marrow-related diseases[23].

## Methods

**Materials.** Cynine5 NHS ester (Cy5) and cyanine7.5 NHS ester (Cy7.5), recombinant mouse protein TNF-α and IL-1β were purchased from Abcam (China), antibodies special for mouse CD36 (anti-rabbit, #18836-1-AP) was obtained from Proteintech (USA), and antibodies special for mouse CD14 (#11390-1), Ki67 (#13030-2), CD31 (#11063-3), MMP9 (#12132), CD68 (#14043),αSMA (#13044), and goat horseradish peroxidase (HRP)-anti-rabbit and goat anti-mouse IgG (#1213 and #1214) were purchased from Servicebio (China), with a dilution ratio of 1:200 for all these antibodies in the immunohistochemical staining. TNF-α ELISA kit, IL-6 ELISA kit, IL-1β ELISA kit, and oxPL-LDL ELISA kit were purchased from Hefei Laier Biotechnology Co., Ltd. (China). Hydrogen peroxide assay kit was supplied by Multi Science (China). Antibodies TNF-R2 (anti-rabbit, #ABP52623) and, CCR-2 (anti-rabbit, #ABP53395) special for mouse were purchased from Abbkine (China), with a dilution ratio of 1:1000 for both antibodies. Antibodies GAPDH (14C10, #5014) and HRP-conjugated anti-rabbit IgG (#7074) were purchased from Cell Signaling Technology (USA), with a dilution ratio of 1:1000 for both antibodies during experimental use. All other chemical reagents were supplied by Sigma Aldrich (USA).

Human umbilical vein endothelial cell line HUVEC and mouse macrophage cell line RAW264.7 were purchased from ATCC (USA), where they were authenticated by cell vitality test, isozyme detection, DNA fingerprinting, and mycoplasma detection. Female ApoE$^{-/-}$ mice (6 weeks) and female C57BL/6 (6 weeks) were purchased from the Animal Facility, Faculty of Health Sciences, University of Macau. All mice were maintained in a dedicated pathogen-free animal facility with free access to food and water in the Institute of Chinese Medical Sciences, University of Macau. All animal procedures were approved by the Animal Ethics Committee, University of Macau, and were compliant with the institutional ethics regulations and guidelines on animal welfare.

**Synthesis and characterization of Oxi-COS.** The synthetic process was according to literature[41]. Briefly, 4-hydroxyphenyl boronic acid pinacol ester (10 mmol) and carbonyldiimidazole (50 mmol) were added to a flame-dried flask with dry dichloromethane (20 mL) under nitrogen. After reaction for 12 h, the raw pinacol boronic ester was washed with H$_2$O (3 × 20 mL), followed by washing with brine (10 mL), dried over MgSO$_4$. The chemical structure of CDI-activated boronic ester carbamate **2** was characterized by NMR spectroscopy. $^1$H NMR (600 MHz, DMSO-d6) of 4-hydroxyphenyl boronic acid pinacol ester (compound **1**): δ 7.65 (d, 2H), 7.35 (d, 2H), 4.79 (s, 2H), 2.5 (s, 1H),1.29 (s, 12H). $^1$H NMR of compound **2**: δ 8.31 (s, 1H), 7.72 (d, 2H), 7.63 (d. 1H), 7.52 (d, 2H), 7.09 (d, 1H), 5.48 (s, 2H), 1.31 (s, 12H).

COS (5 mmol) was dissolved in anhydrous DMSO (10 mL). DMAP (10 mmol) was added, followed by the addition of CDI-activated boronic ester carbamate **2** (10 mmol), and the mixture was shaken on a shaker plate overnight. Subsequently, the modified dextran was dialyzed in dd-H$_2$O to get pure Oxi-COS. The successful conjugation of Oxi-COS was confirmed by NMR spectroscopy. The substitution degree of oxidation-labile compound was quantified by NMR spectroscopy.

**Preparation of ROS responsive NPs and AT-NPs.** Oxi-COS (30 mg) was dissolved in water (deionized, 300 mL) and the mixture was stirred for 1 h. Subsequently, the resulting solution was passed through a syringe filter (with a pore size of 0.45 micron) to have solids removed, and the aqueous solution of NPs was achieved. Upon lyophilization, the resultant powder was put away for use in other experiments. The NPs' morphology was studied by a TEM (H-7650, Hitachi Ltd.) and analyzed by Gatan Digital Micrograph 3.9. The size and size-distribution of the NPs was further characterized via DLS. Subsequently, the NPs were incubated for 48 h in PBS in the presence of a various concentrations of $H_2O_2$ (0, 0.1, 0.2, 0.5, and 1 mM). At predetermined time intervals, the diameters were determined by DLS with a Zetasizer (Nano-ZS, Malvern) system and analyzed by Zetasizer Software (version 7.11).

The preparative process of AT-NPs was similar to that of the micelles as described above. OXi-COS and AT were co-dissolved into deionized water, and the feeding ratio of OXi-COS: AT was 90: 10 (w/w). Upon being stirred for 1 h, the solution was dialyzed with a dialysis bag (WMCO = 12,000) to remove the unencapsulated AT. Subsequently, the solution was subject for filtration and lyophilization for use in the following experiments. The quantity of AT in the AT-NPs was measured by high performance liquid chromatography (HPLC) with reversed-phase column (Agilent TC-C18, $4.6 \times 250$ mm, 5 μm). The mobile phase consisted of ammonium acetate (adjusted pH to 4 with glacial acetic acid) and acetonitrile (6:4, V%), and the detection wavelength was 244 nm. The DEE and DLC were subsequently calculated by using Eqs. (1) and (2), respectively.

$$DEE(\%) = \frac{\text{Mass of drug in NPs}}{\text{Mass of AT in feed}} \times 100\% \qquad (1)$$

$$DLC(\%) = \frac{\text{Mass of drug in NPs}}{\text{Mass of AT} - \text{NPs}} \times 100\% \qquad (2)$$

**Release profile of AT-NPs.** The release profiles of AT from AT-NPs were studied under various concentrations of $H_2O_2$ (0, 0.1, and 1 mM, respectively). AT-NPs (10 mg) dispersed in deionized water (4 mL) were put in a dialysis bag (MWCO = 12,000) that was subsequently placed in 35 mL of buffer solution (sitting in a 37 °C shaker with $30 \times g$ shaking rate). At various time intervals, 2 mL of incubation media was taken out for HPLC analysis, and 2 mL of medium (fresh) was refilled into the buffer solution. The cumulative release rate of AT was calculated using the following equation, Eq. (3).

$$\text{Cumulative drug release}(\%) = \frac{M_t}{M_0} \times 100\% \qquad (3)$$

where $M_t$ was the amount of drug released at time $t$ and $M_0$ was the initial amount of drug in the NPs. The release experiments were conducted in triplicate.

**Isolation of macrophage membrane.** RAW264.7 cells were suspended at a density of $2.0 \times 10^7$ cells $mL^{-1}$ in ice-cold TM buffer solution (pH 7.4; 10 mM Tris + 1 mM $MgCl_2$) and subsequently were extruded through a mini-extruder for at least 30 times in order to disrupt the cells. 1 M sucrose was subsequently mixed with the cell homogenate to eventually reach 0.25 M sucrose concentration, and the mixture was centrifuged at $2000 \times g$ and 4 °C for approximately 10 min. The resulting supernatant was subject for collection upon further centrifugation at $3000 \times g$ for additional 30 min. The cell membranes were collected and washed with ice-cold TM buffer in the presence of 0.25 M of sucrose twice for purification. The bicinchoninic acid assay (BCA) protein assay was employed to analyze the total protein content in the purified macrophage membrane. Approximately, 500 million RAW264.7 cells were able to yield 0.5 mg membrane material (total protein weight). Membrane material was stored at −80 °C for future study.

**Preparation of MM-NPs and MM-AT-NPs.** After isolating the macrophage membrane, the membrane coating was subsequently completed by fusing macrophage membrane vesicles with NPs and AT-NPs (respectively) via a mini-extruder for at least 20 times, and substantial sonication using a bath sonicator at a frequency of 40 kHz and a power of 100 W for 2 min. The resulting solution was centrifuged at $3000 \times g$ for 30 min to remove the uncoated membrane, and was subsequently centrifuged at $12,000 \times g$ for 5 min to remove soluble membrane proteins to obtain the precipitate, crude membrane coated NPs. After washing with PBS for several times until no protein was detected in the supernatant by BCA Protein Assay (Thermo Scientific, Rockford, IL), pure MM-NPs and MM-AT-NPs were therefore obtained. The specific surface markers on macrophage, macrophage membrane and MM-NPs were determined by Western blotting. Membrane protein samples extracted from NPs, macrophage vesicles and MM-NPs were prepared using membrane protein extraction kit (Beyotime, Shanghai, China) according to the manufacturer's protocols. The protein concentrations were determined by the BCA Protein Assay (Thermo Scientific, Rockford, IL). Approximately, 30 μg of protein samples was separated in 10% sodium dodecyl sulfate polyacrylamide gel electrophoresis and electrophoretically transferred onto polyvinylidene fluoride membranes. After blocking with 5% nonfat dry milk in tris-buffered saline, membranes were incubated overnight at 4 °C with primary antibodies, including CD36 (Proteintech), TNFR2 (Abbkine, Inc, China), and CCR2 (Abbkine, Inc., China). Followed by incubation of a secondary HRP–conjugated anti-rabbit IgG

antibody (Cell Signaling Technology) at room temperature for 1 h, specific bands were visualized using an enhanced chemiluminescence detection kit (Bio-Rad).

**In vitro safety evaluation.** In vitro cytotoxicity of free NPs and MM-NPs was evaluated with RAW264.7 and HUVECs cells by MTT assays, respectively. Briefly, the cells were seeded in 96-well plate at a density of $10^5$ per mL. After incubation for 24 h, the media were replaced with fresh media containing NPs or MM-NPs at different concentrations (0.1, 1, 10, and 20 mM), respectively. After incubation for another 24 h, the media were replaced with DMEM containing MTT and the survival number of cells was determined by MTT enzyme-linked immunometric meter (data analyzed by SoftMax Pro 5.4.1). Results were analyzed using GraphPad Prism 6.

**ROS-responsiveness of MM-AT-NPs in vitro.** RAW264.7 cells were incubated with media containing 10 ng $mL^{-1}$ LPS or 20 μg $mL^{-1}$ oxLDL, and concurrently containing MM-AT-NPs at various concentrations of AT (1.25, 5, and 20 μM) for 24 h. The treatment cells with LPS or oxLDL, in the absence and presence of AT or AT-NPs, respectively, was also conducted for comparative purposes. After incubation, the media were replaced with DMEM containing MTT and the survival number of cells was determined by MTT enzyme-linked immunometric meter. Results were analyzed using GraphPad Prism 6.

Furthermore, because LPS is a typical ROS inducer, the ROS levels were analyzed to confirm the ROS production. Briefly, after co-treatment of cells by LPS and the drug (free AT, AT-NPs, and MM-NT-NPs) at 0.4 mM for 12 h, the media were replaced with fresh media containing 2′,7′-dichlorofluorescin diacetate (DCFH-DA). After incubation for 30 min, the cells were washed with PBS for three times and the fluorescence intensity of cells was assessed by flow cytometry (interfaced with BD Accuri C6 Software (version 1. 0. 264. 21)) at an excitation wavelength of 488 nm. Results were analyzed using FlowJo software (version 7.6.1).

In addition to cell viability and ROS production, the apoptosis rate was also measured. After respective co-treatment for 6 h, the cells were suspended in 100 μL of binding buffer, and subsequently mixed with 10 μL of annexin V-fluorescein isothiocyanate (V-FITC) and 10 μL of propidium iodide for 15 min. Another 400 μL of binding buffer was added and the cells were analyzed by a flow cytometer to determine the apoptosis rates. Results were analyzed using FlowJo software (version 7.6.1).

**Cellular uptake and intracellular drug release.** The cellular uptake and intracellular drug release tests were conducted in macrophage, LPS-treated macrophage and foam cells. After seeding the cells and incubation at 37 °C for 24 h, the media were replaced with fresh media containing Cy5-NPs and MM-Cy5-NPs, and the cells were incubated for additional 1–4 h, respectively. Subsequently, the cells were washed for three times with PBS buffer and fixed by paraformaldehyde for 15 min. The cells were washed with PBS for three more times and counterstained with Hoechst for 15 min. After washing with PBS for three times again, the cells were observed via confocal laser scanning microscopy interfaced with LAS X (version 3.5.2) software.

For intracellular drug release behavior, NR, another fluorescence probe, was loaded into the NPs (NR-NPs) and MM-NPs (MM-NR-NPs). NR has no fluorescence itself, but it would show strong red fluorescence once contacting with the lipid droplet inside cells. LPS treated RAW264.7 cells and foam cells were incubated in the media containing NR-NPs and MM-NR-NPs, respectively. The cells were subsequently imaged via microscopy with an excitation wavelength at 480 nm after continuous incubation for 1, 2, 4, and 8 h, respectively.

**Safety evaluation of MM-NPs i.v. injected in mouse.** To evaluate the biocompatibility of our proposed formulations (macrophage membrane coated ROS-responsive NPs) in vivo, 2 groups of 6-week-old female C57BL/6 mice ($n = 6$ in each group) were i.v. injected with saline and MM-NPs, respectively, for half a month with a high dose of 100 mg/kg MM-NPs every 4 days. The changes of bodyweight of mice in both groups were recorded during the 15-day follow-up study. At the end of experiment, all mice were sacrificed, and the heart, livers, spleen, lungs, and kidneys were collected for histological study. In addition, the blood serum was collected for blood chemistry analysis, including the liver function biomarkers (ALT and AST) and kidney function biomarkers (BUN and UA) analysis, as well as inflammatory cytokines (TNF-α, IL-6, and IL-1β) analysis.

**Safety evaluation of macrophage i.v. injected in mouse.** To evaluate the immune response and organ toxicity of foreign RAW264.7 cells in mice, female ApoE$^{-/-}$ mice were randomly and investigator-blindly divided into 4 group ($n = 6$), and intravenously administered with 100 μL of macrophages ($2.5 \times 10^8$, $5 \times 10^8$, and $1 \times 10^9$ macrophages $kg^{-1}$, respectively). The mice were injected once every four days for 2 weeks. Their body weight and survival rate were recorded. At the endpoint of experiment, all mice were sacrificed and the blood was collected to quantitate the immune-associated cells. In addition, their livers, spleen, lungs, and kidneys were processed for histological examination.

**Construction of atherosclerosis in ApoE$^{-/-}$ mice**. Six-week-old female ApoE$^{-/-}$ mice were given high fat diet containing 21.2% lard, 49.1% carbohydrate, 19.8% protein, and 0.2% cholesterol for 3 months, in order to induce atherosclerosis. At the end of treatment, six mice were euthanized and the degree of pathological changes were evaluated by measuring the lesion area of the aorta from the heart to the iliac bifurcation. To determine the extent of atherosclerosis at the aortic root, aortic arch, and brachiocephalic artery, Oil Red O (ORO) staining was performed to confirm the formation of atherosclerotic plaque in mice.

**In vivo targeting of the atherosclerotic plaque**. Six-week-old female ApoE$^{-/-}$ mice fed with high fat food (with the same composition as described above) for 1 month were i.v. administered with Cy7.5, Cy7.5-NPs, MM-Cy7.5-NPs, and Cy7.5-NPs/MAs with the same dosage of Cy7.5 (2 mg kg$^{-1}$), respectively, via the tail vein. After allowing these agents to distribute for 6 h, the mice were euthanized and subsequently perfused with PBS to remove unbound dyes. The aortas were isolated for imaging and quantitative analysis using an IVIS (Lumina XR III), with an excitation wavelength of $780 \pm 20$ nm and an emission wavelength of $840 \pm 20$ nm for the measurements.

**In vivo therapeutic efficacy study**. Six-week-old female ApoE$^{-/-}$ mice were randomly and investigator-blindly divided into 5 groups ($n = 13$, the first batch of 6 and second batch of 7), including a control group (saline), a free-AT treated group, and groups separately treated with AT-NPs, MM-AT-NPs, and AT-NPs/MAs. The mice were treated with high-fat food (with the same composition as described above) for 3 months in a row. After treatment with high-fat diet for the first month, five groups were i.v. administered with saline, AT, AT-NPs, MM-AT-NPs, and AT-NPs/MAs, respectively at a dosage of 2 mg kg$^{-1}$ AT (except for the saline-treated group) per week for additional 2 months. After administration for 2 months, all mice were euthanized and atherosclerotic plaques from ten mice in each group were collected for ORO staining. Imaging of atherosclerotic plaques were conducted for evaluating the therapeutic efficacy of these different formulations. Furthermore, quantitative analysis of atherosclerotic plaque was also determined by Image-Pro Plus 6.0.

**Histological study on aorta tissues**. For histological analysis, aortic root, aortic arch and brachiocephalic artery from mice with various treatments were collected for hematoxylin–eosin (H&E) staining and Masson's trichrome staining. For immunohistochemistry analysis, sections of aortic root, aortic arch and brachiocephalic artery were incubated with antibodies, including CD68, CD14, MMP-9, and $\alpha$-SMA, CD31 and KI67, respectively.

**Quantitation of cytokines, chemokines, and cholesterol**. In addition, the tissues were homogenized for analysis of inflammatory cytokines and chemokines including IL-1$\beta$, IL6, and oxPL-LDL by Elisa kits from Hefei Laier Biotechnology Co., Ltd. (China), including IL-1$\beta$ Elisa kit (catalog number: LE-M0444), IL6 Elisa kit (catalog number: LE-M0458), and oxPL-LDL Elisa kit (catalog number: LE-M1000). Briefly described here, 50 $\mu$L of standard was added to a standard well, and 10 $\mu$L of testing sample and 40 $\mu$L of kit diluent was added to a testing well. Subsequently, 100 $\mu$L of HRP-conjugate reagent was added to each well, followed by coverage with an adhesive strip. After incubation for 60 min at 37 °C, each well was washed and any remained solution in the well should be remove completely. Subsequently, 50 $\mu$L of chromogen solution A and 50 $\mu$L of chromogen solution B were added to each well. After incubation for 15 min at 37 °C in dark, 50 $\mu$L of Stop Solution was added to each well. Finally, optical density (O.D.) at 450 nm was measured by using a microtiter plate reader immediately. The H$_2$O$_2$ levels were determined by using fluorimetric hydrogen peroxide assay kit. The serum from these groups were also collected and the cytokines including TNF-$\alpha$, IL6, and oxPL-LDL were quantified by using Elisa kits according to the previously described procedure. TC and HDL-C contents in serums were quantified using assay kits from Nanjing Jiancheng Bioengineering Institute (China), including TC assay kit (catalog number: A111-1-1) and HDL cholesterol assay kit (catalog number: A112-1-1). Non-HDL-C was subsequently calculated from "TC minus HDL-C".

**DHE staining analysis of aorta tissues**. The samples of aorta root, aorta arch, and brachiocephalic artery were embedded in Tissue-Tek O.T.C. compound, and then 8 $\mu$m sections were incubated with 2% Triton X-100 for 10 min at 21 °C, followed by blocking with 5% BSA in PBS. Subsequently, the sections were stained by DHE (Servicebio, Wuhan, China) and incubated for 30 min under dark environment. After washing with PBS for three times, the slides were imaged under fluorescence microscopy. The fluorescent intensity was quantified by Image-Pro Plus 6.0.

**Binding analysis of cytokine and chemokine by MM-NPs**. TNF-$\alpha$ (10 ng mL$^{-1}$), IL-1$\beta$ (10 ng mL$^{-1}$), oxLDL (20 $\mu$g mL$^{-1}$), and MCP-1 (10 ng mL$^{-1}$) were mixed with MM-NPs at the concentration from 0 to 10 mg mL$^{-1}$. The mixture was incubated at 37 °C for 2 h and then centrifuged at $13,000 \times g$ for 10 min to remove the NPs. The supernatant was analyzed by Elisa kit for the measurements of TNF-$\alpha$, IL-1$\beta$, MCP-1, and oxPL-LDL. The results were further analyzed by GraphPad Prism 6.

**Oil Red O staining in macrophage**. Macrophages were co-treated with oxLDL and different doses of MM-NPs (0, 1.25, 2.5, 5, and 10 mg mL$^{-1}$) for 24 h, and the cells were washed by PBS for three times. Subsequently, macrophages were stained by freshly prepared ORO working solution for 15 min, and rinsed with 60% iso-propanol. After washing with PBS for three times again, the nuclei of macrophages were slightly stained with alum haematoxylin. Finally, macrophages were washed with PBS and measured under microscope.

**Statistical analysis**. One-way ANOVA and two-way was utilized for statistical analysis. Value of $^{*}P \leq 0.05$, $^{**}P \leq 0.01$ and $^{***}P \leq 0.001$ were applied to annotate statistical significance. All data were presented as mean value $\pm$ the standard deviation of independent experiments.

**Reporting summary**. Further information on research design is available in the Nature Research Reporting Summary linked to this article.

## Data availability

The source data underlying Figs. 1c, d, g–i, 2a, c–f, 3c, f, 4a, b–i, 5b, c–d, f, and Supplementary Figs. 3a–c, 4a–c, f, h, 6, 8b–d, 9a, b, 10–13, and 14a–e are provided as a Source Data file. The data supporting all the plots within this paper are available from the corresponding authors upon request.

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

## Acknowledgements

Science and Technology Development Fund (FDCT), Macao SAR (Grant no. 0121/2018/A3), National Science Foundation of China (Grant no. 21871301), and Research Committee at University of Macau (Grant nos. MYRG2017-00010-ICMS and MYRG2016-00165-ICMS-QRCM) are gratefully acknowledged for providing financial support to this work.

## Author contributions

The project was conceptually designed by C.G., S.L., and R.W. The majority of the experiments were conducted by C.G., Q.H., and C.L. (with equal contributions), assisted by C.K. and L.Y. Data analysis and interpretation were done by C.G., J.W., S.L., and R.W. The paper was prepared by C.G. and R.W. All authors discussed the results and implications, and commented on the paper.

## Competing interests

The authors declare no competing interests.
