## [Peer Review File · Nature Communications]

Reviewers' Comments:

Reviewer #1:

Remarks to the Author:

Although the study contains some interesting elements, its speculative nature and unsubstantiated statements, dampen this reviewer's enthusiasm of the work. The title is misleading. What do the authors consider 'chemotherapy' and what involves 'immunosuppression'?

- Conceptual comment on nanoparticle platform: When the macrophage coating is applied, how do the authors ensure that all NPs are coated and/or how is excess coating removed? In the Methods section, no purification process is included, which is inappropriate as the type of coating process described will require extensive washing steps.

The physicochemical characterization of MM-NP is limited at best. TEM images are of poor quality and should be evaluated quantitatively. The authors need to demonstrate that excess macrophage coating is not present.

H₂O₂ experiments should not merely be performed on AT-NPs, but also on the nanoparticles that are coated with a macrophage membrane, MM-AT-NP.

The same for AT release, it should also be studied in the presence of the macrophage membrane.

- In vitro studies: The authors suggest therapeutic experiments are conducted on both LPS and oxLDL created macrophages: "Attenuation effects on LPS-induced inflammation of macrophages and oxLDL-induced formation of foam cells". This appears not to be the case. The therapeutic experiments were conducted on LPS-treated cells.

The induction of apoptosis is not desired. Macrophage apoptosis is a hallmark of plaque rupture risk.

How were the NPs labeled with Cy5? Was Cy5 conjugated to one of the components or was it simply mixed in?

The in vitro experiments presented in Figure 2g are by no means quantitative and no attempt were made to analyze the data. The "Trojan horse" disguise is pure speculation and needs to be much better substantiated.

The same applies to the Nile red experiments and ROS responsiveness. This section is highly speculative, only derivative evidence is presented.

Conceptual comment on in vivo therapy experiments. The RAW264.7 cell line is not ideal, not for the experiments described above, but certainly not for therapeutic experiments in mice. The RAW264.7 cell line was established from a tumor induced by the Abelson murine leukemia virus. The experiments should be conducted with bone marrow derived macrophages or a cell line with a high degree of similarity with plaque macrophages. also, group sizes of are too small, and the main text reported group size (n=6) is different from the figure legend (n=3).

How were the different NPs (Cy7.5-NP, MM-Cy7.5-NP and Cy7.5-NP/MAs dosed? Based on what parameter? Protein, Cy7.5, amount of NPs?

The blood half-lives of Cy7.5-NP, MM-Cy7.5-NP and Cy7.5-NP/MAs should have been determined first. This parameter dictates the imaging time point, which –based in this reviewer's experience– is too short at 6 hours.

The in vitro experiments presented in Figure 2g are by no means quantitative and no attempt were made to analyze the data. The "Trojan horse" disguise is pure speculation and needs to be much better substantiated.

What filter set was used for IVIS imaging?

NIRF imaging is not a quantitative technique and therefore comments like the one below should not be included.

"The enhanced targeting efficiency of Cy7.5-NPs/MAs was likely attributed to activated immune response involving inflammatory signal pathways by live macrophages, although both macrophage membrane and live macrophage have surface adhesion capacity through the interactions between

membrane proteins and inflammatory cytokines.”

The data and analyses presented in Figure 3f need to be evaluated by a statistician. AT should not be given i.v., but given orally.

The data and analyses presented in Figure 3f need to be evaluated by a statistician.

Additional major comments:

Group sizes of the data presented in Figure 4 are too small.

The section “Anti-atherosclerotic mechanism of MM-NPs” is highly speculative.

Reviewer #2:

Remarks to the Author:

Previous studies have shown that cell membrane coated nanoparticles (NPs) from both red blood cells (RBC) and platelets, and neutrophil membranes can alleviate inflammatory arthritis. It has been shown that atorvastatin (AT) inhibits plaque development and adventitial neovascularization in ApoE^{-/-} mice. In this manuscript, the authors synthesized and characterized the RAW264.7 cells-derived membrane-atorvastatin coated on the surface of ROS responsive nanoparticles (MM-AT-NPs) via an extrusion method and sought to investigate: 1) the effect of atorvastatin loaded ROS-NPs (AT-NPs) and macrophage membrane coated AT-NPs (MM-AT-NPs) on lipopolysaccharide (LPS) and oxidative low-density lipoprotein (oxLDL) induced macrophage inflammation and foam cell formation; and 2) the effect of the targeting efficiency and therapeutic efficacy of MM-AT-NPs and AT-NPs internalized inside macrophages (AT-NPs/MAs) in an atherosclerotic model ApoE^{-/-} mice. The novel findings of this study include 1) macrophage membrane improves targeted delivery of NPs and payload to the lesion site, acts as a scavenger for proinflammatory factors to suppress the immune response; 2) the MM-AT-NPs, rather than AT-NPs/MAs, exhibited higher therapeutic efficacy to treat the inflammatory disease. This is an interesting study and the findings are intriguing. However, there are some issues that may limit the strength of the conclusions, which need to be addressed.

Major Comments:

1. With regard to the efficacy of MM-NP on atherosclerosis in Figure 3, the number of mice (n=3 aorta tissues) used in each group of this study is too low. To obtain the valid atherosclerotic plaque data, the minimal of mice should be 8 in each group or more (many studies use n = 10-20). Please include the description on how the atherosclerosis plaque sizes were analyzed. (Recommendation on Design, Execution, and Reporting of Animal Atherosclerosis Studies: A Scientific Statement From the American Heart Association. ATVB. 2017)
2. The authors only showed the representative ORO of the proximal aortic lesions (Sup Fig. s8), and did not provide quantified results of the proximal aortic lesions from different groups in this study. It is of importance to quantify the proximal aortic lesion size with ORO staining from ApoE^{-/-} mice. 10- 15 sections should be used to analyze the plaque size reliably in each mouse. (Recommendation on Design, Execution, and Reporting of Animal Atherosclerosis Studies: A Scientific Statement From the American Heart Association. ATVB. 2017)
3. It has been shown that atorvastatin inhibits plaque development in ApoE deficient mice without affecting total cholesterol levels (Atorvastatin inhibits plaque development and adventitial neovascularization in ApoE deficient mice independent of plasma cholesterol levels. Atherosclerosis. 2011). Does AT-NPs change the body weight and the cholesterol levels in this study,
4. AT has been reported to reduce atherosclerosis through inhibition of endothelial proliferation

and reducing the number of perivascular CD31(+) neovessels, thereby inhibiting adventitial neovascularization (Atherosclerosis 2011. PMID: 21130458). It is not clear whether the impact of MM-AT-NPs is mediated by endothelial cells or adventitial neovascularization?

5. Labeling the nanoparticle with Cy7.5 is to image inflammation in vivo. Topical application of PMA onto the ear lobes of mice induces acute inflammation, manifested by local swelling, erythema and infiltration of immune cells. So it would be important to verify whether the probe is able to detect the superficial inflammation induced by PMA in mice. (Bioluminescence imaging of myeloperoxidase activity in vivo. Nature medicine. 2009. PMID: PMC2831476).

6. It would be important to examine whether the probe fluorescence is dependent on the activity of myeloperoxidase (MPO) in mice embedded with MPO and glucose oxidase matrigel. (Bioluminescence imaging of myeloperoxidase activity in vivo. Nature medicine. 2009. PMID: PMC2831476).

7. It would be important to look at infiltration of macrophages or macrophage composition in the proximal aortic lesion in this study.

8. Figure S9, it is concluded that MM-AT-NPs ameliorated plaque in an atherosclerotic mouse model by only showing representative photographs of brachiocephalic artery sections stained by H&E, CD14 antibody, MMP-9 antibody, Masson's trichrome and α -SMA antibody. These results need to be quantified to conclude MM-AT-NPs ameliorated plaque in an atherosclerotic mouse model.

9. Supplementary Fig S6, the labeling of figures and the scale bar are not correct. It seems the labeling on the figure is mirrored.

10. A scale bar should be added in Fig 4, Supplementary Fig. S4 and Supplementary Fig. S5.

11. Description of statistical analysis should be included in the methods of this study, including the normality test, analysis of student's t test or One-way ANOVA.

Minor:

Page 4, "a novel delivery system derived from macrophage cell membrane coated ROS responsive NPs (MM-NPs) for the treatment of atherosclerosis" should be "a novel delivery system derived from macrophage membrane coated ROS responsive NPs (MM-NPs) for the treatment of atherosclerosis"

Page 10, Figure3 figure legend, "ORO stained aorta tissues collected form atherosclerotic mice after treatment with various formulations." Please correct the typo.

Reviewer #3:

Remarks to the Author:

In the article "Targeted chemotherapy and immunosuppression of atherosclerosis by macrophage-biomimetic ROS-responsive nanoparticles," the authors describe the preparation of a ROS-responsive, macrophage-membrane cloaked nanoparticle prepared using chitosan-oligosaccharide, which is subsequently used to deliver an atorvastatin for atherosclerosis treatment. The authors performed an interesting comparison between cell membrane-coated nanoparticles and nanoparticles loaded inside live macrophages, which represent two very popular bio-inspired drug delivery strategies in the field. The authors also reach the conclusion that in the present case, cell membrane coated nanoparticles perform better than nanoparticle-loaded macrophages in treating atherosclerosis, which is attributed to the absence of inflammatory signaling in the cell membrane coated nanoparticles. There is a recently published study demonstrating the use of red blood cell-

membrane coated nanoparticle for atherosclerosis management (Wang et al. *Advanced Sciences* doi: 10.1002/advs.201900172). The authors should cite the study to highlight the new design principles in the present work. The comparison between membrane-coated nanoparticles and nanoparticle-loaded macrophages can be interesting. However, further characterizations and platform justifications are necessary before the article can be recommended to the broad audience of *Nature Communications*. These aspects are as follows:

1. The reviewer struggles to find information of drug loading yield and efficiency in the article, despite that there is description for drug loading calculation in the Method section. The authors should report the drug loading/efficiency or perhaps place the information somewhere more obvious.
2. The H₂O₂-mediated release was performed entirely with non-membrane coated nanoparticles. However, since cell membrane coating has been observed to influence particle stability and drug release, the effect of H₂O₂ should be examined on macrophage-coated nanoparticles. For instance, would the nanoparticles still aggregate in the presence of H₂O₂ following membrane coating? Can H₂O₂ still access the oxidation-sensitive nanoparticle core? These are critical questions that should be discussed with proper experimentation.
3. H₂O₂ concentrations ranging from 0.01mM to 1mM were used in characterizing the nanoparticles. The typical concentration range of H₂O₂ at atherosclerotic sites should be referenced to give proper context to the experimental design.
4. There is no characterization of AT nanoparticle-loaded macrophages. Information regarding these macrophages, including viability, nanoparticle and drug loading efficiency, as well as other targeting-related functional studies are critical for proper assessment of the work.
5. For the atherosclerotic plaque assessment (Fig. 3f) and inflammatory response characterization (Fig 4b-g), it is very surprising to the reviewer that the authors are able to obtain statistically significant data with a small sample size (n=3). These biological phenomena are usually highly variant. Despite the low sample size, the representative pictures posted by the authors do not correlate with the reported data on the first glance. For instance, the authors struggle to see differences among different groups in Fig 3e. In addition, Fig 4a is difficult to decipher. The authors should include proper visual aids (i.e. annotations, arrows, insets) to highlight the area of interest in these figures.
6. The authors make strong assertion in the result and discussion section claiming the membrane coated nanoparticles provide better treatment efficacy than nanoparticle-loaded macrophages despite that live macrophage can better target atherosclerotic sites. However, most of the comparative results have no statistical significance between these two groups (i.e. Fig 3c, Fig 3f, Fig 4b-g). At best the authors may suggest that the membrane coated nanoparticles show hints of improved treatment efficacy based on the given experimental data. Some of the current assertions in the manuscript, however, are way too strong of a statement to be justified.
7. The article should provide more discussion of atherosclerosis treatment. For instance, does the present approach slows down the atherosclerosis progression or does it help with plaque resolution? Such discussion is important in justifying the treatment design.
8. Several landmark papers on cell membrane coated nanoparticle platform should be referenced, including the first report of macrophage membrane coated particle (*Nat Nanotechnol.* 2013 Jan;8(1):61-8.) and the first report of cell membrane coated nanoparticle for cardiovascular disease treatment (*Nature* 526, pages118–121, 2015).
9. ROS and H₂O₂ are not “expressed,” they are “produced” in inflammatory tissues. Many instances of ROS/H₂O₂ “overexpression” in the article should be corrected.

Response Letter

Reviewer #1

Although the study contains some interesting elements, its speculative nature and unsubstantiated statements, dampen this reviewer's enthusiasm of the work. The title is misleading. What do the authors consider 'chemotherapy' and what involves 'immunosuppression'?

Response: We are grateful to this reviewer for the critical comments about some of our speculative statements as well as the specific comment on our manuscript title, which we have seriously considered and addressed. The term "chemotherapy" was originally used as a generic term in a wider context to refer to therapy by a chemical species (drug molecule). The term "immunosuppression" was referring to the sequestration of proinflammatory cytokines by macrophage membrane, as the process involves "suppression" of inflammation and potentially further immune response. To address this critics and to avoid speculation, the title has now been changed to a more objective description, "*Treatment of atherosclerosis by macrophage-biomimetic ROS-responsive nanoparticles via targeted pharmacotherapy and sequestration of proinflammatory cytokines*".

- Conceptual comment on nanoparticle platform: When the macrophage coating is applied, how do the authors ensure that all NPs are coated and/or how is excess coating removed? In the Methods section, no purification process is included, which is inappropriate as the type of coating process described will require extensive washing steps. The physicochemical characterization of MM-NP is limited at best. TEM images are of poor quality and should be evaluated quantitatively. The authors need to demonstrate that excess macrophage coating is not present.

Response: Thanks for pointing out that the purification process was not described for MM-NPs preparation in Methods section. The excess membrane was removed by centrifugation at a low speed (3000 \square g). The centrifugation speed was subsequently increased to 12,000 \square g to remove soluble membrane proteins in the supernatant and to precipitate MM-NPs. The obtained MM-NPs were further washed by PBS and precipitated by centrifuge for several cycles until no protein was detected in the supernatant by BCA Protein Assay (excess macrophage coating is not present any more). The purification process has now been added into the Methods section.

To ensure that the macrophage coating was properly applied to all NPs, the macrophage membrane coated NPs was carefully characterized by TEM, DLS and western blot analysis. To address "TEM images are of poor quality", we have repeated TEM characterizations of these NPs, and the TEM images of MM-NPs were updated with higher quality. In addition, all NPs were quantitatively analyzed by DLS. TEM image (**Fig. 1b** and **1f**) showed that nearly all NPs were coated with membrane, as the thickness of wrapped layer was ~ 9 nm, in line with the thickness of cell membrane. DLS (**Fig. 1h**) further confirmed that the diameter of NPs was increased from ~ 204 nm to ~ 227 nm, likely due to the addition of cell membrane layers, and the zeta potential of membrane coated NPs was also similar to the macrophage membrane (**Fig. 1g**). Additionally, western blot analysis (**Fig. 1i**) confirmed the presence of key macrophage membrane antigens on the surfaces of MM-NPs, such as

TNFR, CD36 and CCR2, suggesting that the macrophage membrane was successfully coated onto NPs. We have updated this part of discussions in the revised manuscript.

H₂O₂ experiments should not merely be performed on AT-NPs, but also on the nanoparticles that are coated with a macrophage membrane, MM-AT-NP.

The same for AT release, it should also be studied in the presence of the macrophage membrane.

Response: Thanks for these important advices. Accordingly, we have conducted the experiment on the responsiveness of MM-NPs towards H₂O₂. As shown in **Supplementary Fig. S3b** in the revised version, after incubation for up to 100 min, the hydrolysis rate reached ~ 75%. The hydrolysis rate of MM-NPs was moderately slower than that of AT-NPs, likely attributed to the barrier effects of the coated membrane. In addition, AT release from MM-AT-NPs was also studied, with the total AT release percentage reaching up to ~74.6%, in response to 1 mM of H₂O₂ (**Supplementary Fig. S3c**), moderately less than that of AT-NPs (~ 80%), suggesting that the membrane coating minimally affected H₂O₂ responsiveness of the NPs.

- In vitro studies: The authors suggest therapeutic experiments are conducted on both LPS and oxLDL created macrophages: “Attenuation effects on LPS-induced inflammation of macrophages and oxLDL-induced formation of foam cells”. This appears not to be the case. The therapeutic experiments were conducted on LPS-treated cells.

Response: Thanks for the comment. In this part of our studies, the cell viability, ROS generation and apoptosis experiments were performed to evaluate the therapeutic effects of MM-AT-NPs on LPS-treated macrophages (**Figure 2**). In addition, we actually found that MM-AT-NPs also improved the cell viability of oxLDL treated macrophages (**Supplementary Fig. S4c**) and prevented the formation of oxLDL-induced immature dendritic cells-like morphological changes of macrophage (**Supplementary Fig. S5**). Although the relevant dataset on the therapeutic effects of MM-AT-NPs are placed in the Supplementary Information, we have made brief discussion in the main text.

The induction of apoptosis is not desired. Macrophage apoptosis is a hallmark of plaque rupture risk.

Response: We agree with the reviewer’s opinion that macrophage apoptosis is not desired in the plaques development. Our studies did show that all AT formulations inhibited cellular apoptosis induced by LPS treatment. As this is not a key action mechanism, we have now moved the dataset into Supplementary Information (**Supplementary Figure S4f-g**).

How were the NPs labeled with Cy5? Was Cy5 conjugated to one of the components or was it simply mixed in?

Response: Cy5 NHS ester, being lipophilic, was loaded into the hydrophobic core of NPs during the self-assembly process of the amphiphilic polysaccharides (during the formation of NPs), similar to the way that AT was loaded. To add clarity, we added brief description about the process of NPs labelling (loading) with Cy5 under

“Cellular uptake and intracellular drug release” in Results section.

The *in vitro* experiments presented in Figure 2g are by no means quantitative and no attempt were made to analyze the data. The “Trojan horse” disguise is pure speculation and needs to be much better substantiated. The same applies to the Nile red experiments and ROS responsiveness. This section is highly speculative, only derivative evidence is presented.

Response: Thanks for your comments. To address this critics, quantitative analysis on the cellular uptake and intracellular drug release was conducted via flow cytometry, and the results were updated into the revised **Figure 2e** and **2f**. In addition, to avoid speculations, we deleted the term “Trojan horse” and related descriptions. We have also significantly revised the discussion about Nile red (NR) uptake experiments, in order to be more accurate and objective in the description of the results, without any extended scientific speculation.

Conceptual comment on *in vivo* therapy experiments. The RAW264.7 cell line is not ideal, not for the experiments described above, but certainly not for therapeutic experiments in mice. The RAW264.7 cell line was established from a tumor induced by the Abelson murine leukemia virus. The experiments should be conducted with bone marrow derived macrophages or a cell line with a high degree of similarity with plaque macrophages. also, group sizes of are too small, and the main text reported group size (n=6) is different from the figure legend (n=3).

Response: Thanks for the suggestion on the use of more suitable macrophages in the experiments. We agree with the reviewer that bone marrow derived macrophages would be an ideal cell line in our study. However, we humbly believe that RAW264.7 cell line serves the purpose of our study as a model macrophage cell line, and our results (both *in vitro* and *in vivo*) have indeed supported the validity of this model. More importantly, in several important pioneering works about macrophage-biomimetic drug delivery, the membranes of RAW264.7 cells were used to coat various NPs or used directly as live cells based drug carriers, which have demonstrated high targeted delivery efficiency to various inflammatory diseases as well as decent therapeutic efficacy, including rheumatoid arthritis (*Nano Lett.*, 2019, 19, 124-134), lung metastasis of breast cancer (*ACS Nano* 2016, 10, 7738-7748), and primary glioblastoma (*Advanced Materials* 2015, 4, 1645-1652). To properly credit the reviewer’s comment, we have added a statement in Discussion section that “bone marrow derived macrophages or a cell line with a high degree of similarity with plaque macrophages might act as a better macrophage model and source of membrane for biomimetic drug delivery systems to target atherosclerotic lesion.”

In the anti-atherosclerosis *in vivo* experiment, each group had six mice (n = 6) and three of them (n = 3) were used for ORO staining on aorta tissues. The rest of them were utilized for other characterizations and analyses. To further increase the sample size, we conducted additional anti-atherosclerosis experiments and extended the number of mice to 10 for each group for ORO staining studies (n = 10, three from the first batch and seven from the second batch). The updated results were shown in the updated **Figure 3e**.

How were the different NPs (Cy7.5-NP, MM-Cy7.5-NP and Cy7.5-NP/MAs dosed? Based on what parameter? Protein, Cy7.5, amount of NPs?

Response: The doses of these dye-loaded NPs (Cy7.5-NPs, MM-Cy7.5-NPs and Cy7.5-NPs/MAs) were based on the amount of free Cy7.5 (e.g. 2 mg/kg), prepared from the same batch of Cy7.5-NPs to ensure consistency. To add clarity, we had added the dose information in the sections of Methods and Results.

The blood half-lives of Cy7.5-NP, MM-Cy7.5-NP and Cy7.5-NP/MAs should have been determined first. This parameter dictates the imaging time point, which –based in this reviewer’s experience– is too short at 6 hours.

Response: Thanks for the constructive suggestion. To have better understanding about half-lives of these NPs, the pharmacokinetic profiles of Cy7.5-NPs, MM-Cy7.5-NPs and Cy7.5-NPs/MAs were studied and the result was shown in **Supplementary Fig. S10** in the revised manuscript. After *i.v.* administration of these formulations into C57BL/6 mice, the blood was collected at different time points (up to 24 h), and placed in 96-well plate with an equal volume. The fluorescence intensity of Cy7.5 in these blood samples was subsequently measured by IVIS spectrum system. As shown in **Supplementary Fig. 10**, the fluorescence intensities of all groups were significantly decreased since the 6 h time point after administration, and became significantly weaker at 12 h and 24 h time points. Based on these results, the plaque targeting studies at 6 h time point seems appropriate, and the results did demonstrate the significantly different fluorescence intensities in the aorta tissues among different groups, and support our conclusions about the different targeting efficiencies of those formulations.

What filter set was used for IVIS imaging?

NIRF imaging is not a quantitative technique and therefore comments like the one below should not be included. “The enhanced targeting efficiency of Cy7.5-NPs/MAs was likely attributed to activated immune response involving inflammatory signal pathways by live macrophages, although both macrophage membrane and live macrophage have surface adhesion capacity through the interactions between membrane proteins and inflammatory cytokines.”

Response: The filter used for IVIS imaging (Fig. 3b) was set with an exciting wavelength of 780 ± 20 nm and an emission wavelength of 840 ± 20 nm, the detail of which was added into the revision. To comply with the reviewer’s kind suggestion, we deleted the above-mentioned sentence.

The data and analyses presented in Figure 3f need to be evaluated by a statistician.

Response: Quantitative analysis on aortic pathologies was conducted with the standard protocol outlined in “Atherosclerosis - En Face Aorta and Analysis” (Daugherty Lab, Saha Cardiovascular Research Center, University of Kentucky), which has also been described in “Recommendation on Design, Execution, and Reporting of Animal Atherosclerosis Studies: A Scientific Statement From the American Heart Association”, *Arter. Thromb. Vasc. Biol.*, 2017, 37(9), 131-157). The data was already verified by a second researcher who was blinded to the experimental groups. And to fulfill the requirement by the reviewer, data analysis of **Figure 3f** was independently verified by a statistician from the Institute of Chinese Medical Sciences, University of Macau.

AT should not be given i.v., but given orally.

Response: For the clinical prevention and management of cardiovascular disease, statins are mainly used as an oral formulation. However, AT has also been used in nanoformulations to effectively regress atherosclerotic plaques, via *i.v.* administration (*Journal of Controlled Release*, 2018, 283, 241-260). Herein, AT was employed in our study as a model drug to demonstrate the potential of our macrophage-biomimetic nanoparticles for targeted therapy of atherosclerosis (free AT was extensively used for comparison in our *in vivo* evaluations).

Additional major comments:

Group sizes of the data presented in Figure 4 are too small.

Response: Additional experiments were conducted and the number of mice in **Figure 4** has been extended to 6-7 per group (n = 6-7). Accordingly, **Figure 4** has been updated.

The section “Anti-atherosclerotic mechanism of MM-NPs” is highly speculative.

Response: Thanks for this comment about potentially speculative discussions. We have revised the section to make all discussions more objective, straightforward, and reflective of the observations.

Reviewer #2

Previous studies have shown that cell membrane coated nanoparticles (NPs) from both red blood cells (RBC) and platelets, and neutrophil membranes can alleviate inflammatory arthritis. It has been shown that atorvastatin (AT) inhibits plaque development and adventitial neovascularization in Apoe^{-/-} mice. In this manuscript, the authors synthesized and characterized the RAW264.7 cells-derived membrane-atorvastatin coated on the surface of ROS responsive nanoparticles (MM-AT-NPs) via an extrusion method and sought to investigate: 1) the effect of atorvastatin loaded ROS-NPs (AT-NPs) and macrophage membrane coated AT-NPs (MM-AT-NPs) on lipopolysaccharide (LPS) and oxidative low-density lipoprotein (oxLDL) induced macrophage inflammation and foam cell formation; and 2) the effect of the targeting efficiency and therapeutic efficacy of MM-AT-NPs and AT-NPs internalized inside macrophages (AT-NPs/MAs) in an atherosclerotic model Apoe^{-/-} mice. The novel findings of this study include 1) macrophage membrane improves targeted delivery of NPs and payload to the lesion site, acts as a scavenger for proinflammatory factors to suppress the immune response; 2) the MM-AT-NPs, rather than AT-NPs/MAs, exhibited higher therapeutic efficacy to treat the inflammatory disease. This is an interesting study and the findings are intriguing.

Response: We are extremely grateful to the reviewer for the kind summary of the main findings and the novelty of our work.

However, there are some issues that may limit the strength of the conclusions, which need to be addressed.

Major Comments:

1. With regard to the efficacy of MM-NP on atherosclerosis in Figure 3, the number of mice (n=3 aorta tissues) used in each group of this study is too low. To obtain the valid atherosclerotic plaque data, the minimal of mice should be 8 in each group or more (many studies use n = 10-20). Please include the description on how the atherosclerosis plaque sizes were analyzed. (Recommendation on Design, Execution, and Reporting of Animal Atherosclerosis Studies: A Scientific Statement From the American Heart Association. *ATVB*. 2017).

Response: Thanks for your comment on the sample size. We conducted additional anti-atherosclerotic experiments in mice and extended the total number of mice to 10 for each group in Figure 3 (n = 10, three from the first batch and seven from the second batch). The revised results were shown in Figure 3e (updated). In addition, atherosclerotic lesions were determined by following en face analysis of lesions on the intimal surface of the aorta (“Atherosclerosis - En Face Aorta and Analysis”, Daugherty Lab, Saha Cardiovascular Research Center, University of Kentucky), which has also been described in “Recommendation on Design, Execution, and Reporting of Animal Atherosclerosis Studies: A Scientific Statement From the American Heart Association,” *Arter. Thromb. Vasc. Biol.*, 2017, 37(9), 131-157), and the data analysis was verified by a second researcher who was blinded to the experimental groups. With the reviewer’s reminder, we have now included the description of how the plaques sizes were analyzed in the section of “Therapeutic efficacy in atherosclerotic mouse”.

2. The authors only showed the representative ORO of the proximal aortic lesions (Sup Fig. S8), and did not provide quantified results of the proximal aortic lesions from different groups in this study. It is of importance to quantify the proximal aortic lesion size with ORO staining from Apoeddf mice. 10- 15 sections should be used to analyze the plaque size reliably in each mouse. (Recommendation on Design, Execution, and Reporting of Animal Atherosclerosis Studies: A Scientific Statement From the American Heart Association. *ATVB*. 2017).

Response: Thanks for the important suggestions. Based on this advice and also recommendation from the literature (Recommendation on Design, Execution, and Reporting of Animal Atherosclerosis Studies: A Scientific Statement From the American Heart Association, *Arter. Thromb. Vasc. Biol.*, 2017, 37(9), 131-157), sequential 10 sections (10 μm thickness) at 100 μm intervals in the atherosclerosis-susceptible region of the aortic root were acquired for each mouse in the study. The results were shown in **Supplementary Fig. S11**, and further quantitative analysis on lesion sizes was conducted via Image Pro.

3. It has been shown that atorvastatin inhibits plaque development in ApoE deficient mice without affecting total cholesterol levels (Atorvastatin inhibits plaque development and adventitial neovascularization in ApoE deficient mice independent of plasma cholesterol levels. *Atherosclerosis*. 2011). Does AT-NPs change the body weight and the cholesterol levels in this study.

Response: Thanks for this insightful question. In order to answer this question, we carefully evaluated the changes of body weight in mice treated with AT, AT-NPs, MM-AT-NPs and AT-NPs/MAs, respectively. The results showed that these formulations had negligible effects on the body weight changes of the mice, as shown by the new data set in **Fig. 4i**. In addition, the total cholesterol (TC), low density lipoprotein cholesterol (LDL-C), and high density lipoprotein cholesterol (HDL-C) were all determined in this batch of studies. As shown in **Fig. 4h**, the TC level and LDL-C level changed little in the MM-AT-NPs treated mice (consistent with the literature report), while HDL-C level was increased very moderately in all the treated groups.

4. AT has been reported to reduce atherosclerosis through inhibition of endothelial proliferation and reducing the number of perivascular CD31(+) neovessels, thereby inhibiting adventitial neovascularization (Atherosclerosis 2011. PMID: 21130458). It is not clear whether the impact of MM-AT-NPs is mediated by endothelial cells or adventitial neovascularization?

Response: Thanks for the insightful comments on the potential impact of MM-AT-NPs on endothelial cells and adventitial neovascularization. In order to reveal this potential action mechanism of MM-AT-NPs, CD31 immunohistochemical study and KI67 staining analysis were conducted in sections of the aorta root, with the new set of data added into updated **Figure 4a**. Quantitative analysis showed that the number of perivascular CD31(+) neovessels was reduced and endothelial proliferation (as observed from KI67 staining analysis) was also inhibited, consistent with the previous literature report (*Atherosclerosis*, 2011, 214(2), 295-300). Thus, the inhibition of adventitial neovascularization was indeed involved in the anti-atherosclerosis mechanism of MM-AT-NPs. These results and discussions have been added into the revised manuscript.

5. Labeling the nanoparticle with Cy7.5 is to image inflammation in vivo. Topical application of PMA onto the ear lobes of mice induces acute inflammation, manifested by local swelling, erythema and infiltration of immune cells. So it would be important to verify whether the probe is able to detect the superficial inflammation induced by PMA in mice. (Bioluminescence imaging of myeloperoxidase activity in vivo. *Nature medicine*. 2009. PMID: PMC2831476).

Response: Sorry for the confusion caused. Cyanine7.5 (Cy7.5) itself cannot target or detect inflammation. It is just a near infrared (NIR) dye with long-wave emission that allows deep tissue penetration. In our studies, we employed Cy7.5 as an imageable payload (loaded into the core of NPs) to track the bio-distribution of NPs, MM-NPs and NPs/MAs in atherosclerotic mice. We have made minor revision in the manuscript to add more clarity about the role of Cy7.

6. It would be important to examine whether the probe fluorescence is dependent on the activity of myeloperoxidase (MPO) in mice embedded with MPO and glucose oxidase matrigel. (Bioluminescence imaging of myeloperoxidase activity in vivo. *Nature medicine*. 2009. PMID: PMC2831476).

Response: We apologize for the confusion caused. Cy7.5 is a synthetic dye belonging to polymethine group, used in our study as an imageable, trackable NIR dye. It is

relatively stable without any responsiveness or specificity towards inflammation (*Nano Today*, 2018, 18, 124-136). It has been used in many studies as a reliable probe to track biodistribution of various nanomaterials *in vivo* (e.g. *Advanced Functional Materials*, 2013, 24 (17), 2450-2461 & *Biomaterials*, 2019, 213, 119219).

7. It would be important to look at infiltration of macrophages or macrophage composition in the proximal aortic lesion in this study.

Response: Thanks for the comment. We conducted CD14 immunohistochemical staining on the aorta root to reveal the macrophage composition (**Figure 4a**). As shown by the dataset, the quantitative analysis of CD14 stained sections indicated that the content of macrophage in the aorta roots was reduced in mice treated with different AT formulations, in comparison with that of the control group. Relevant discussion has been added into the revised manuscript.

8. Figure S9, it is concluded that MM-AT-NPs ameliorated plaque in an atherosclerotic mouse model by only showing representative photographs of brachiocephalic artery sections stained by H&E, CD14 antibody, MMP-9 antibody, Masson's trichrome and α -SMA antibody. These results need to be quantified to conclude MM-AT-NPs ameliorated plaque in an atherosclerotic mouse model.

Response: Thanks a lot for this constructive suggestion. To fulfill this requirement, the results of aorta root sections stained by H&E, CD14 antibody, MMP-9 antibody, Masson's trichrome, α -SMA antibody, CD31 antibody and KI67 antibody have all been quantitatively analyzed and summarized in **Fig. 4a**. Collectively, these results indicated that MM-AT-NPs efficiently ameliorated plaque in an atherosclerotic mouse model.

9. Supplementary Fig S6, the labeling of figures and the scale bar are not correct. It seems the labeling on the figure is mirrored.

Response: Thanks for your kind attentions on this error on this Figure (renamed as **Supplementary Fig. S7** in revised version due to addition of other figures). This figure labeling and scale bar has been corrected.

10. A scale bar should be added in Fig 4, Supplementary Fig. S4 and Supplementary Fig. S5.

Response: Scale bar in **Fig. 4a**, Supplementary Fig. S4 (renamed as **Supplementary Fig. S5** in the revised version), and Supplementary Fig. S5 (renamed as **Supplementary Fig. S6** in the revised version) has been added.

11. Description of statistical analysis should be included in the methods of this study, including the normality test, analysis of student's t test or One-way ANOVA.

Response: Thanks for this kind reminder. The section of "Statistical analysis" was added in Methods section.

Minor:

Page 4, "a novel delivery system derived from macrophage cell membrane coated

ROS responsive NPs (MM-NPs) for the treatment of atherosclerosis” should be “a novel delivery system derived from macrophage membrane coated ROS responsive NPs (MM-NPs) for the treatment of atherosclerosis”

Response: Corrected as suggested.

Page 10, Figure3 figure legend, “ORO stained aorta tissues collected form atherosclerotic mice after treatment with various formulations.” Please correct the typo.

Response: Thanks for the kind correction. The typo “form” was corrected to “from”.

Reviewer #3

In the article “Targeted chemotherapy and immunosuppression of atherosclerosis by macrophage-biomimetic ROS-responsive nanoparticles,” the authors describe the preparation of a ROS-responsive, macrophage-membrane cloaked nanoparticle prepared using chitosan-oligosaccharide, which is subsequently used to deliver an atorvastatin for atherosclerosis treatment. The authors performed an interesting comparison between cell membrane-coated nanoparticles and nanoparticles loaded inside live macrophages, which represent two very popular bio-inspired drug delivery strategies in the field. The authors also reach the conclusion that in the present case, cell membrane coated nanoparticles perform better than nanoparticle-loaded macrophages in treating atherosclerosis, which is attributed to the absence of inflammatory signaling in the cell membrane coated nanoparticles. There is a recently published study demonstrating the use of red blood cell-membrane coated nanoparticle for atherosclerosis management (Wang et al. *Advanced Sciences* doi: 10.1002/advs.201900172). The authors should cite the study to highlight the new design principles in the present work. The comparison between membrane-coated nanoparticles and nanoparticle-loaded macrophages can be interesting.

Response: We are grateful to this reviewer for the kind comments of our work. The highly related reference about red blood cell-membrane coated nanoparticle for atherosclerosis management (Wang et al. *Advanced Sciences* doi: 10.1002/advs.201900172) has been introduced briefly and referenced in Introduction.

However, further characterizations and platform justifications are necessary before the article can be recommended to the broad audience of *Nature Communications*. These aspects are as follows:

1. The reviewer struggles to find information of drug loading yield and efficiency in the article, despite that there is description for drug loading calculation in the Method section. The authors should report the drug loading/efficiency or perhaps place the information somewhere more obvious.

Response: Thanks a lot for the kind reminder. The drug encapsulation efficiency and drug loading content of AT-NPs were actually calculated already, it was our negligence that we did not put the data into the manuscript. They were 48.3% and 5.1%, respectively. This information has now been added into the section of

“Preparation and characterization of ROS responsive NPs and MM-NPs”.

2. The H₂O₂-mediated release was performed entirely with non-membrane coated nanoparticles. However, since cell membrane coating has been observed to influence particle stability and drug release, the effect of H₂O₂ should be examined on macrophage-coated nanoparticles. For instance, would the nanoparticles still aggregate in the presence of H₂O₂ following membrane coating? Can H₂O₂ still access the oxidation-sensitive nanoparticle core? These are critical questions that should be discussed with proper experimentation.

Response: Thanks for the insightful questions (same with Reviewer 1’s comment). The stability of MM-AT-NPs was measured in PBS with and without H₂O₂, respectively. As shown in **Supplementary Fig. S3a**, MM-AT-NPs was rather stable in PBS. As expected, the hydrolysis rate of MM-NPs was increased in response the increased concentration of H₂O₂ (**Supplementary Fig. S3b**), moderately slower than that of NPs without MM coating. Accordingly, the drug release profile of MM-AT-NPs was also determined in PBS with and without H₂O₂, respectively (**Supplementary Fig. S3c**): MM-AT-NPs exhibited an accumulative release rate of ~74.6% in response to 1 mM of H₂O₂, which was slightly lower AT-NPs that exhibited an accumulative AT release rate of ~80%. Thus, there results indicated that the effect of macrophage membrane coating on the H₂O₂ responsiveness of AT-NPs was rather minimal.

3. H₂O₂ concentrations ranging from 0.01 mM to 1 mM were used in characterizing the nanoparticles. The typical concentration range of H₂O₂ at atherosclerotic sites should be referenced to give proper context to the experimental design.

Response: The exact ROS level in atherosclerotic site has rarely been reported due to the complicated components of ROS species (including H₂O₂, superoxide, hydroxyl radical, singlet oxygen, and alpha-oxygen) as well as different atherosclerotic degrees in different aorta site (*Arterioscler Thromb Vasc Biol.* 2017, 37, 41-52). While the level of H₂O₂ we measured in the plaque tissues was 0.2-0.4 mM, we should note that H₂O₂ is just one component of ROS species. In a few important literature papers that reported the use of ROS responsive NPs for targeted treatment of atherosclerosis (e.g. *Journal of the American College of Cardiology*, 2018, 72 (21), 2591-2605 & *ACS Nano*, 2018, 12, 8943-8960), the ROS responsiveness of NPs were initially studied in H₂O₂ form 0.01 mM to 1 mM as representative ROS conditions. Thus we selected the same concentration range of H₂O₂ in the present study. In Results section, these relevant discussions were added.

4. There is no characterization of AT nanoparticle-loaded macrophages. Information regarding these macrophages, including viability, nanoparticle and drug loading efficiency, as well as other targeting-related functional studies are critical for proper assessment of the work.

Response: Thanks for the important comment. The cell viability of macrophages internalized with AT-NPs was therefore determined via MTT assays, and the results showed a safe profile of AT-NPs towards macrophage even at the highest concentration of AT at 20 μM (**Supplementary Fig. S9b**). The internalization of NPs into macrophages was also studied by incubating macrophage with cell culture

medium containing Cy5-NPs for 2 h, and the macrophages were subsequently imaged by laser scanning confocal microscope to show clear evidence of internalization (Supplementary Fig. 9a). The content of AT in cells (5×10^7) internalized with AT-NPs was determined by HPLC, and the drug loading efficiency of AT-NPs/MAs was ~14.7%. Regarding their plaque targeting efficiency, Cy7.5-NPs/MAs exhibited significant accumulation in the plaques (Fig. 3b), indicating that the inflammatory tropism of NPs internalized macrophage was well maintained. Relevant discussions have been added into the revised version.

5. For the atherosclerotic plaque assessment (Fig. 3f) and inflammatory response characterization (Fig 4b-g), it is very surprising to the reviewer that the authors are able to obtain statistically significant data with a small sample size (n=3). These biological phenomena are usually highly variant. Despite the low sample size, the representative pictures posted by the authors do not correlate with the reported data on the first glance. For instance, the authors struggle to see differences among different groups in Fig 3e. In addition, Fig 4a is difficult to decipher. The authors should include proper visual aids (i.e. annotations, arrows, insets) to highlight the area of interest in these figures.

Response: Thanks for this important comments, which are consistent with relevant comments from the other two reviewers. As mentioned already in the responses to the other two reviewers, we conducted additional batch of therapeutic experiments on atherosclerotic mice, and the number of mice was extended to 10 per group for plaque assessment (n = 10, three from the first batch and seven from the second batch). The results were shown in the updated Fig. 3e. Atherosclerotic lesions shown in Fig. 3f were determined by using en face analysis of lesions on the intimal surface of the aorta (Recommendation on Design, Execution, and Reporting of Animal Atherosclerosis Studies: A Scientific Statement From the American Heart Association, *Arter. Thromb. Vasc. Biol.*, 2017, 37(9), 131-157), and the measurements were verified by a second researcher who was blinded to the experimental groups. All the analytical data were independently verified by a statistician. In addition, arrows and insets were added in Fig. 3e and Fig. 4a to highlight the area of interest, as suggested by the reviewer.

6. The authors make strong assertion in the result and discussion section claiming the membrane coated nanoparticles provide better treatment efficacy than nanoparticle-loaded macrophages despite that live macrophage can better target atherosclerotic sites. However, most of the comparative results have no statistical significance between these two groups (i.e. Fig 3c, Fig 3f, Fig 4b-g). At best the authors may suggest that the membrane coated nanoparticles show hints of improved treatment efficacy based on the given experimental data. Some of the current assertions in the manuscript, however, are way too strong of a statement to be justified.

Response: We are very grateful to this reviewer for the very kind suggestions about the strength of our statement. Accordingly, we have revised the section of Results and Discussion, and concluded that membrane coated nanoparticles showed hints of better therapeutic efficacy against atherosclerotic mice, when compared to nanoparticle-loaded macrophages.

7. The article should provide more discussion of atherosclerosis treatment. For instance, does the present approach slows down the atherosclerosis progression or does it help with plaque resolution? Such discussion is important in justifying the treatment design.

Response: Thanks for this important suggestion. Based on all of the datasets (including those that have been added upon addressing all of the reviewers' comments), the therapeutic effects of MM-AT-NPs seem to mainly act on atherosclerosis progression. We have made substantial discussions in Discussion section about this.

8. Several landmark papers on cell membrane coated nanoparticle platform should be referenced, including the first report of macrophage membrane coated particle (Nat Nanotechnol. 2013 Jan;8(1):61-8.) and the first report of cell membrane coated nanoparticle for cardiovascular disease treatment (Nature 526, pages118-121, 2015).

Response: Thanks for the kind reminder. These two highly relevant papers are now introduced briefly and referenced in Introduction.

9. ROS and H₂O₂ are not “expressed,” they are “produced” in inflammatory tissues. Many instances of ROS/H₂O₂ “overexpression” in the article should be corrected.

Response: Thanks for the kind correction. All relevant words, such as “expressed”, “overexpression” or “over-expressed” related to ROS/H₂O₂, have been changed into “produced”, “overproduction” or “over-produced”.

Reviewers' Comments:

Reviewer #2:

Remarks to the Author:

It has been shown previously that atorvastatin (AT) inhibits plaque development and adventitial neovascularization in Apoe^{-/-} mice. This study examined the impact of macrophage membrane coated ROS-responsive nanoparticles (NPs) to coat AT on atherosclerosis in Apoe^{-/-} mice. The authors have responded in detail to the Reviewers comments and many of the concerns have been addressed. However, a number of major concerns remain.

1. The authors have increased the number of mice per group for the atherosclerosis studies from 3 to 10 by adding a separate study of 7 mice per group. "The Apoe^{-/-} mice were randomly and investigator-blindly divided into 5 groups (n = 13, the first batch of 6 and second batch of 7), including a control group (saline)," It would be very important to provide the gender of all the mice used, especially for atherosclerosis study. Were male or female mice used or both? There are well described gender effects on atherosclerosis in mice. In the absence of this information, the results are difficult to interpret.

2. A previous study has shown atorvastatin reduces HDL-C in plasma of mice (Statins increase hepatic cholesterol synthesis and stimulate fecal cholesterol elimination in mice. *J Lipid Res.* 2016 Aug; 57(8): 1455–1464. doi: 10.1194/jlr.M067488). Therefore, it is surprising to see atorvastatin (AT) can increase HDL-C by 50% (Fig 5h) without affecting total cholesterol and LDL-C levels in Apoe^{-/-} mice in this study. It would be important to include how the levels of HDL-C and LDL-C were measured. It would be important to examine the distribution of cholesterol in the lipoprotein fractions using FPLC, which is widely used in mouse studies to define changes in lipoprotein levels. Furthermore, Apoe^{-/-} mice accumulate remnant lipoproteins not LDL-C in their blood. It would be better to use Non-HDL-C (TC-HDL) than LDL-C as a measure of apoB containing lipoproteins.

3. Figure 4. The quality of the immunocytochemistry staining appears to be poor, especially for CD31 and Ki67, where it is difficult to appreciate the positive staining. This raises concerns about the reliability of the quantification of this data. The figure is too crowded making it hard to interpret. The quantified data, including the proximal atherosclerotic lesion area, macrophage content in the lesions, MMP, and collagen content are too small to read. It would be better to re-organize the figures.

4. Page 15: "ApoE^{-/-} mice were given normal diet containing 0.2% cholesterol and 21% lard for 3 months, in order to induce atherosclerosis". Page 16: "ApoE^{-/-} mice fed with high fat food for 1 month were i.v. administered with Cy7.5". Were all of these mice on the same high fat diet? Please provide the composition of the high fat diet used in this study in more detail. It is confusing to describe a diet containing 21% lard as normal. Please describe the types of diet used more clearly.

5. With regard to measurement of oxLDL by ELISA, which type of oxidized LDL was measured exactly, HNE, MDA, or oxPL-LDL? It would be important to provide this information.

6. "In addition, the tissues were homogenized for analysis of inflammatory cytokines and chemokines including IL-1 β , IL6 and oxLDL by Elisa kit." Please provide the information (procedures, company name, catalog number) on ELISA.

7. "Viability of RAW264.7 co-treated with LPS (10 ng mL⁻¹) and AT, AT-NPs and MM-AT-NPs, respectively, at 0, 1.25 μ M, 5 μ M and 20 μ M AT. b. c, Intracellular ROS levels (by flow cytometry analysis) in RAW264.7 cells treated with LPS (400 ng mL⁻¹)," the concentration of LPS for the viability study of RAW is 10ng/mL, but the concentration of LPS was 400ng/mL for the Intracellular ROS levels study. Initial cell populations were gated for a live population using FSC and SSC plot of cell only sample. The gate was set to remove cell debris and dead cells (small FSC and SSC).

400ng/mL of LPS is a high dosage. As shown in FigS4f, g, high dosage of LPS causes a large amount of macrophage death. Therefore, it would be important to provide a representative figure of the gating strategy use for the ROS release experiment.

8. Fig S4g shows the quantified result from FigS4f, the apoptotic rate is 20.8% from AT-NPs+LPS, 18.6% from MM-AT-NPs+LPS in FigS4f, but there is no this data in FigS4g. Was the data normalized to something else?

9. Which regression model was used in Figure1C&D, and Figure5b? It seems it is not linear regression in Figure 5b. Please include the statistical approach for these results.

Minor:

There are a lot of typographical and grammatical errors, especially involving special signs or units, please read it carefully and correct them on page 13 and page14. Typo: Page 13: "bu_er solution (pH 7.4; 10 mM Tris + 1 mM MgCl₂) and subsequently were extruded" Figure 1 figure legend "macrophages (2.5_10⁷ cells)".

Reviewer #3:

Remarks to the Author:

The authors have made substantial improvements to the manuscript by increasing the subject number in critical experiments, performing rigorous statistical analysis, and providing additional experimental characterizations to address potential concerns. My prior comments have all been addressed and the revised manuscript is recommended for publication.

Reviewer #4:

Remarks to the Author:

In this revised manuscript, Gao et al report a novel approach where they developed an ROS-responsive coated with a macrophage derived layer. These nanoparticles were found to be quite effective as a treatment for atherosclerosis when loaded with atorvastatin.

Overall, the work seems to be well-done and fairly comprehensive. I have a few suggestions for improvement prior to publication:

-The authors should add data that speaks to the in vivo biocompatibility of their proposed treatment. Examples might include histology, clinical blood chemistry, etc.

-The authors should calculate and add the circulation times of their agents.

-The work should be compared with that of Willem Mulder on simvastatin loaded biomimetic HDL nanoparticles.

-Figure 1 g, h, j – statistical comparisons should be made.

Response Letter

Reviewer #2

It has been shown previously that atorvastatin (AT) inhibits plaque development and adventitial neovascularization in Apoe^{-/-} mice. This study examined the impact of macrophage membrane coated ROS-responsive nanoparticles (NPs) to coat AT on atherosclerosis in Apoe^{-/-} mice. The authors have responded in detail to the Reviewers comments and many of the concerns have been addressed. However, a number of major concerns remain.

Response: We are very grateful to this reviewer for recognizing the efforts that we have made to address all the comments from the first round of review, as well as for the new comments raised during this round of review, which has helped again to further improve the quality of the work.

1. The authors have increased the number of mice per group for the atherosclerosis studies from 3 to 10 by adding a separate study of 7 mice per group. “The Apoe^{-/-} mice were randomly and investigator-blindly divided into 5 groups (n = 13, the first batch of 6 and second batch of 7), including a control group (saline),” It would be very important to provide the gender of all the mice used, especially for atherosclerosis study. Were male or female mice used or both? There are well described gender effects on atherosclerosis in mice. In the absence of this information, the results are difficult to interpret.

Response: Thanks for this important reminder. The mice used in our experiment were all female, and the information was just added into the main text of the manuscript.

2. A previous study has shown atorvastatin reduces HDL-C in plasma of mice (Statins increase hepatic cholesterol synthesis and stimulate fecal cholesterol elimination in mice. *J Lipid Res.* 2016 Aug; 57(8): 1455–1464. doi: 10.1194/jlr.M067488). Therefore, it is surprising to see atorvastatin (AT) can increase HDL-C by 50% (Fig 5h) without affecting total cholesterol and LDL-C levels in Apoe^{-/-} mice in this study. It would be important to include how the levels of HDL-C and LDL-C were measured. It would be important to examine the distribution of cholesterol in the lipoprotein fractions using FPLC, which is widely used in mouse studies to define changes in lipoprotein levels. Furthermore, Apoe^{-/-} mice accumulate remnant lipoproteins not LDL-C in their blood. It would be better to use Non-HDL-C (TC-HDL) than LDL-C as a measure of apoB containing lipoproteins.

Response: Thanks for these insightful comments. We have also noticed in several references that atorvastatin could reduce HDL-C level in plasma of ApoE^{-/-} mice after long-term treatment (e.g. *J. Lipid Res.*, 2016, 57(8), 1455-1464; *Eur. Heart J.*, 2015, 36(1), 39-48). However, in some other references (e.g. *Biomed. Pharmacother.*, 2019, 109, 1445-1453; *Exp. Ther. Med.*, 2018, 16(5), 3785-3792), we found that atorvastatin could increase HDL-C level by 50 %-90 % in plasma of ApoE^{-/-} mice, similar to what was observed in our study. Other statin drugs, such as simvastatin and pravastatin, were

also found to increase HDL-C level (e.g. *Arterioscl. Throm. Vas.*, 2005, 25, 1426-1432; *Lipids Health Dis.*, 2011, 10, 8).

We also agree with the reviewer that FPLC would be a better tool to measure the individual lipoprotein levels. As we do not have access to this equipment, the lipoprotein levels in the plasma of ApoE^{-/-} mice were measured using assay kits including Total Cholesterol Assay Kit, Low-density Lipoprotein Cholesterol Assay Kit and High-density Lipoprotein Cholesterol Assay Kit (all of which were supplied by Nanjing Jiancheng Bioengineering Institute), which have also been commonly employed to measure lipoprotein fraction levels (e.g. *Hepatology*, 2016, 63(4), 1190-1204; *Biomedicine & Pharmacotherapy*, 2019, 113, 108753; *Diabetes Care*, 2010, 33(7), 1625-1628). The details of the measurements on TC and HDL-C levels were added into the manuscript (Page 17 and 18 in the track-changes manuscript). And to follow this reviewer's advice, Non-HDL-C was calculated from "TC minus HDL-C" and the data was updated in revised Fig. 4h by replacing LDL-C with Non-HDL-C.

3. Figure 4. The quality of the immunocytochemistry staining appears to be poor, especially for CD31 and Ki67, where it is difficult to appreciate the positive staining. This raises concerns about the reliability of the quantification of this data. The figure is too crowded making it hard to interpret. The quantified data, including the proximal atherosclerotic lesion area, macrophage content in the lesions, MMP, and collagen content are too small to read. It would be better to re-organize the figures.

Response: Thanks for your kind suggestions. In order to improve the legibility of those images, the immunocytochemistry staining images of CD31 and Ki67 were amplified and moved to **Supplementary Fig. S13**. The rest of the histological analysis and immunohistochemical analysis of aorta root sections, as well as the quantitative analysis in **Figure 4**, have been re-organized to improve legibility and clarity.

4. Page 15: "ApoE^{-/-} mice were given normal diet containing 0.2% cholesterol and 21% lard for 3 months, in order to induce atherosclerosis". Page 16: "ApoE^{-/-} mice fed with high fat food for 1 month were i.v. administered with Cy7.5". Were all of these mice on the same high fat diet? Please provide the composition of the high fat diet used in this study in more detail. It is confusing to describe a diet containing 21% lard as normal. Please describe the types of diet used more clearly.

Response: Thanks a lot for pointing this typo out. The word "normal" was accidentally inserted there. In both cases, ApoE^{-/-} mice were fed with high fat diet. We have revised the sentence accordingly (in the updated Page 16). The composition of high fat diet used in this study was "21.2% lard, 49.1% carbohydrate, 19.8% protein and 0.2% cholesterol". This detailed composition was added in the manuscript (Page 16).

5. With regard to measurement of oxLDL by ELISA, which type of oxidized LDL was measured exactly, HNE, MDA, or oxPL-LDL? It would be important to provide this information.

Response: The ox-LDL Elisa assay kit employed was to selectively measure oxLDL in the phospholipid form (namely, oxPL-LDL). The information was added in the revised manuscript (Page 9).

6. “In addition, the tissues were homogenized for analysis of inflammatory cytokines and chemokines including IL-1 β , IL6 and oxLDL by Elisa kit.” Please provide the information (procedures, company name, catalog number) on ELISA.

Response: The IL-1 β , IL6 and oxLDL Elisa kits were purchased from Hefei Laier Biotechnology Co., Ltd., China, and the catalog number were LE-M0444, LE-M0458 and LE-M1000, respectively. This information was added in Method section (Page 17). In addition, the analytical procedures of IL-1 β , IL6 and oxLDL were also added in the manuscript (Page 17).

7. “Viability of RAW264.7 co-treated with LPS (10 ng mL⁻¹) and AT, AT-NPs and MM-AT-NPs, respectively, at 0, 1.25 μ M, 5 μ M and 20 μ M AT. b. c, Intracellular ROS levels (by flow cytometry analysis) in RAW264.7 cells treated with LPS (400 ng mL⁻¹),” the concentration of LPS for the viability study of RAW is 10ng/mL, but the concentration of LPS was 400ng/mL for the Intracellular ROS levels study. Initial cell populations were gated for a live population using FSC and SSC plot of cell only sample. The gate was set to remove cell debris and dead cells (small FSC and SSC). 400ng/mL of LPS is a high dosage. As shown in FigS4f, g, high dosage of LPS causes a large amount of macrophage death. Therefore, it would be important to provide a representative figure of the gating strategy use for the ROS release experiment.

Response: Thanks for the reminder. Yes, 400 ng/mL of LPS indeed induced relatively high level of cell death on RAW264.7 cells. Accordingly, we added gating strategy used for the ROS-inducing experiment on RAW264.7 cells, and the representative figures of gating strategy were shown in **Supplementary Fig. S4d**.

8. Fig S4g shows the quantified result from FigS4f, the apoptotic rate is 20.8% from AT-NPs+LPS, 18.6% from MM-AT-NPs+LPS in FigS4f, but there is no this data in FigS4g. Was the data normalized to something else?

Response: Thanks for the questions. Actually, the apoptotic rates of 20.8% and 18.6%, respectively from AT-NPs+LPS and MM-AT-NPs+LPS treated groups, were both late apoptotic rates (Q2 in **Supplementary Fig. S4g** (with updated figure sequence due to SI revision)). However, **Supplementary Fig. S4h** (with updated sequence) showed the results of total apoptosis rates in all groups, i.e. the sum of early apoptotic rate (Q3) and late apoptotic rate (Q2) in **Supplementary Fig. S4g** (updated sequence), and the total apoptotic rates were 36.6% and 24.26%, respectively for AT-NPs+LPS and MM-AT-NPs+LPS treated groups.

9. Which regression model was used in Figure1C&D, and Figure5b? It seems it is not linear regression in Figure 5b. Please include the statistical approach for these results.

Response: Nonlinear regression fitting with inhibitory dose-response model (variable slope model) was employed to process data in **Fig. 5b** using Graphpad Prism 6. The information was added in figure legends of **Fig. 5b** (Page 22). **Fig. 1C** and **Fig. 1D** are the cumulative hydrolysis rate and cumulative drug release curve, respectively, and all the data points were connected with error bars (standard deviation) from three repeats.

Minor:

There are a lot of typographical and grammatical errors, especially involving special

signs or units, please read it carefully and correct them on page 13 and page 14. Typo: Page 13: “bu_er solution (pH 7.4; 10 mM Tris + 1 mM MgCl₂) and subsequently were extruded”

Figure 1 figure legend “macrophages (2.5_10⁷ cells)”.

Response: Thanks for your kind advice. We have found that some of these typos incurred during PDF conversion. We have carefully gone through the manuscript several times to correct typos and to improve the overall writing quality.

Reviewer #3

The authors have made substantial improvements to the manuscript by increasing the subject number in critical experiments, performing rigorous statistical analysis, and providing additional experimental characterizations to address potential concerns. My prior comments have all been addressed and the revised manuscript is recommended for publication.

Response: We appreciate the reviewer for recognizing our tremendous efforts in previous round of revisions and for recommending our manuscript for publication.

Reviewer #4

In this revised manuscript, Gao et al report a novel approach where they developed an ROS-responsive coated with a macrophage derived layer. These nanoparticles were found to be quite effective as a treatment for atherosclerosis when loaded with atorvastatin.

Overall, the work seems to be well-done and fairly comprehensive. I have a few suggestions for improvement prior to publication:

Response: We are grateful to this reviewer for the kind comments on our previously revised work.

-The authors should add data that speaks to the *in vivo* biocompatibility of their proposed treatment. Examples might include histology, clinical blood chemistry, etc.

Response: Thanks for your kind suggestion. In order to systemically evaluate the biocompatibility of our proposed formulations (macrophage membrane coated ROS-responsive NPs, aka MM-NPs), we conducted the following experiment: two groups of 6-week female C57BL/6 mice (n = 6 in each group) were *i.v.* injected with saline and MM-NPs, respectively, for half a month with a high dose of 100 mg/kg MM-NPs every four days. At the end of the experiment, the heart, livers, spleens, lungs and kidneys from these mice were collected for histological study, and the results (**Supplementary Fig. S8a**) showed that no obvious damage was observed in mice treated with high-dose MM-NPs. In addition, blood serum was also collected for clinical chemistry analysis. As shown in **Supplementary Fig. S8b**, the liver function biomarkers (alanine transaminase (ALT) and aspartate aminotransferase (AST)) and kidney function biomarkers (blood urea nitrogen (BUN) and uric acid (UA)) in the serum of mice treated with MM-NPs were comparable to those of the saline-treated group.

Furthermore, as shown in **Supplementary Fig. S8c**, the inflammatory cytokine levels (TNF- α , IL-6 and IL-1 β) in the serum of mice treated with MM-NPs showed no obviously changes in comparison to those of the saline-treated group. Meanwhile, the changes of bodyweight of both the control group and treatment group were also similar (**Supplementary Fig. S8d**). Collectively, this new set of data support the good biocompatibility profile of our formulation. These data are briefly discussed in the updated manuscript (Page 6).

-The authors should calculate and add the circulation times of their agents.

Response: Cyanine 7.5 NHS ester (Cy7.5), a synthetic dye with near infrared emission (excitation/emission 788/808 nm), was employed as a trackable payload in the forms of Cy7.5-loaded NPs (Cy7.5-NPs), macrophage membrane coated Cy7.5-NPs (MM-Cy7.5-NPs), and Cy7.5-NPs internalized inside macrophage (Cy7.5-NPs/MAs) in order to track the *in vivo* pharmacokinetics of these formulations in a mouse model (**Supplementary Fig. S11**). The circulation half-life ($t_{1/2}$) values of these formulations were added in the revised manuscript (Page 7). MM-Cy7.5-NPs ($t_{1/2}$ = 9.82 h) exhibited longer circulation time than that of Cy7.5-NPs ($t_{1/2}$ = 5.43 h). And Cy7.5-NPs/MAs ($t_{1/2}$ = 13.32 h) had highest circulation time.

-The work should be compared with that of Willem Mulder on simvastatin loaded biomimetic HDL nanoparticles.

Response: Thanks for the advice. Accordingly, this important work by Prof. Mulder (*Nat. Commun.* 2014; 5: 3065) has been cited in the revised manuscript and our platform was briefly compared with simvastatin loaded biomimetic HDL nanoparticles in the Discussion section (Page 11 and 12 in the track-changes manuscript).

-Figure 1 g, h, j – statistical comparisons should be made.

Response: Thanks for the kind reminder. The statistical comparisons have been made in **Fig. 1g, h and j**.

Reviewers' Comments:

Reviewer #2:

Remarks to the Author:

The authors have adequately addressed all of the Reviewers' comments with one minor exception.

Minor:

Comment #5: "The ox-LDL Elisa assay kit employed was to selectively measure oxLDL in the phospholipid form (namely, oxPL-LDL). The information was added in the revised manuscript (Page 9)." To be accurate, please replace the ox-LDL with oxPL-LDL in the main text, Figure 4g and the figure legend.

Reviewer #4:

Remarks to the Author:

The authors have addressed my concerns.

Response to Reviews

Reviewer #2

The authors have adequately addressed all of the Reviewers' comments with one minor exception.

Minor:

Comment #5: "The ox-LDL Elisa assay kit employed was to selectively measure oxLDL in the phospholipid form (namely, oxPL-LDL). The information was added in the revised manuscript (Page 9)." To be accurate, please replace the ox-LDL with oxPL-LDL in the main text, Figure 4g and the figure legend.

Response: Thanks for your kind reminder. We have replaced "ox-LDL" (where applicable) with "oxPL-LDL" in the main text. Figure 4g and the corresponding figure legend are also updated with "oxPL-LDL".